# A morphogenetic EphB/EphrinB code controls hepatopancreatic duct formation

M. Ilcim Thestrup[1], Sara Caviglia [1], Jordi Cayuso [2], Ronja L.S. Heyne[1], Racha Ahmad[1], Wolfgang Hofmeister [1,3,4], Letizia Satriano[5], David G. Wilkinson [2], Jesper B. Andersen [5] & Elke A. Ober [1]*

The hepatopancreatic ductal (HPD) system connects the intrahepatic and intrapancreatic ducts to the intestine and ensures the afferent transport of the bile and pancreatic enzymes. Yet the molecular and cellular mechanisms controlling their differentiation and morphogenesis into a functional ductal system are poorly understood. Here, we characterize HPD system morphogenesis by high-resolution microscopy in zebrafish. The HPD system differentiates from a rod of unpolarized cells into mature ducts by de novo lumen formation in a dynamic multi-step process. The remodeling step from multiple nascent lumina into a single lumen requires active cell intercalation and myosin contractility. We identify key functions for EphB/EphrinB signaling in this dynamic remodeling step. Two EphrinB ligands, EphrinB1 and EphrinB2a, and two EphB receptors, EphB3b and EphB4a, control HPD morphogenesis by remodeling individual ductal compartments, and thereby coordinate the morphogenesis of this multi-compartment ductal system.

[1] University of Copenhagen, NNF Center for Stem Cell Biology (DanStem), Blegdamsvej 3B, 2200 Copenhagen N, Denmark. [2] The Francis Crick Institute, Neural Development Laboratory, 1 Midland Road, London NW1 1AT, UK. [3] Odense University Hospital, Laboratory of Molecular and Cellular Cardiology, Department of Clinical Biochemistry and Pharmacology, Sdr. Boulevard 29, 5000 Odense C, Denmark. [4] University of Southern Denmark, Department of Cardiovascular and Renal Research, Winsloewparken 21, 5000 Odense C, Denmark. [5] University of Copenhagen, Biotech Research and Innovation Centre, Copenhagen, Denmark. *email: elke.ober@sund.ku.dk

A critical step in establishing a functional digestive system is the connection of the accessory organs to the intestine. During development, the hepatopancreatic ductal (HPD) system connects the liver, gallbladder (GB) and pancreas to the digestive tract. The HPD system is essential for food processing and uptake of nutrients, as it transports the bile produced in the liver, as well as pancreatic enzymes and electrolytes to the intestine, but little is known about how the HPD develops. It is a multi-component system consisting of the extrahepatic duct (EHD), cystic duct (CD), common bile duct (CBD) and extrapancreatic duct (EPD; Fig. 1f)[1].

A subgroup including EHD, CD and CBD, as well as the gallbladder GB are known as the extrahepatic biliary ducts (EHB). The CBD coming from the liver and the EPD from the pancreas converge into the hepatopancreatic ampulla (HPA), also known as Ampulla of Vater in humans, connecting the HPD system to the intestine. This organization and morphology of the HPD system is conserved between fish and amniotes, however, in mammals the EHDs are commonly referred to as right or left ducts and connecting common duct, and the EPD as pancreatic duct[1,2]. Congenital or environmentally caused malformation of the HPD, such as ductal atresia

**Fig. 1** The HPD system forms by de novo tubulogenesis. **a** The forming HPD is visualized by pan-endodermal *Tg(Xla.Eef1a1:GFP)* (magenta) and ZO-1 (gray) labels the first junctional aggregates in the HPD primordium (blue arrow) at 46 hpf. **b** Ductal endoderm expression of Anxa4 (magenta) and apical aPKC (gray) visualize HPD morphology and nascent microlumina within the prospective CBD and EPD ($n = 17$, $N = 3$). **c** The HPD system elongates and the gallbladder anlage (yellow arrow) becomes detectable at 52 hpf. The immature EHB and HPA lumina show gaps and luminal loops compared to the EPD exhibiting a continuous immature lumen ($n = 21$, $N = 3$). The HPA connects at two points (green arrows). **d** At 60 hpf, all ductal compartments and a sphere-like gallbladder display a continuous emerging lumen, including a single connection of the HPA (green arrow), but excluding the EHD ($n = 29$, $N = 2$). See also Supplementary Movie 1. **e** Along organ growth, the HPD tubes start bending from 72 hpf. The emerging lumen formed in the HPD system is continuous and connected to both IHD and IPD luminal network ($n = 11$, $N = 2$) **f** transgenic *keratin18:GFP* expression visualizes the compact EHB and IHD at 5 dpf ($n = 13$, $N = 4$), see also Supplementary Movie 8. Schematic overviews of HPD differentiation (**a–f**): starting with junctional aggregates and resulting in a continuous lumen spanning the HPD system. From 52 hpf individual HPD compartments are distinguishable: extrahepatic duct (EHD, pink); cystic duct (CD, red); gallbladder (GB, yellow); common bile duct (CBD, orange); extrapancreatic duct (EPD, blue); hepatopancreatic ampulla (HPA, green). Scale bars: **a–f** = 10 μm. **g** Lumen length quantification of the differentiating CBD between 48 and 72 hpf. **h–j** At 5 dpf, cortical actin indicates open lumina in the wild-type EHD, CBD and EPD; scale bars = 10 μm. Magnified cross-sections show open lumina (asterisk) of indicated ductal domains (yellow lines); scale bars = 5 μm. Error bars show SEM; *n* = sample number, *N* = number of experiments. Source data are provided as a Source Data file

caused by duct obstruction or paucity compromise organ function and cause chronic inflammation[3–6]. Pancreatobiliary junction malformations can result in a two-way reflux of bile and pancreatic juice, leading to inflammation and subsequently malignant transformation[7]. This underscores the importance of establishing the ductal compartments and their junctions, however, the causes of congenital HPD malformations are largely unknown.

HPD progenitors arise from an endodermal cell population located between the liver and pancreas anlagen. In mouse, this population co-expresses transcription factors Sox17 and Pdx1, with Sox17 being essential for EHB duct formation[8]. The latter acts in concert with other transcription factors, including Hes1, Hnf6, Hnf1β and Hhex, whose deletion causes HPD malformations ranging from CBD enlargement and GB agenesis to replacement with pancreatic or duodenal cells[9–14]. HPD progenitors in zebrafish are similarly located between the liver and pancreas and express transcription factors Pdx1, Prox1, and Sox9b[14,15]. Intriguingly, lack of Sox9b leads to HPD malformation and ectopically differentiating hepatic and pancreatic cells in the HPD[15,16]. Zebrafish mutants for mesodermal fgf10 exhibit similar cellular plasticity in the HPD, indicating that mesoderm-endoderm interactions are essential for HPD differentiation[2,10]. The morphological emergence of the HPD coincides with the downregulation of Prox1 in HPD progenitors and membrane localisation of calcium binding protein AnnexinA4 (Anxa4)[2,17]. Although the transcriptional network controlling HPD development is emerging[1,18], the morphogenetic events underlying duct differentiation and their molecular regulators are largely unknown. Two classic hypotheses describe the formation of the HPD either by budding and invagination of the foregut epithelium, or by differentiation from a solid tissue (e.g. cord) by subsequent vacuolization and lumen formation. Histological analysis of human embryonic tissue favors that a bile duct lumen buds from the foregut, without transitioning through a solid tissue stage[19]. Due to limited sample accessibility and developmental stages, current understanding is incomplete and an in-depth analysis is necessary.

Eph receptor tyrosine kinases and their cognate Ephrin ligands are regulators of diverse cellular functions, such as cell adhesion, migration and proliferation, which are critical for organ morphogenesis and homeostasis[20–22]. EphB receptors interact primarily with B-type transmembrane Ephrin ligands[21]. Their interaction can uniquely trigger bidirectional signaling upon cell contact, with EphB-expressing cells activating forward signaling and EphrinB-bearing cells eliciting reverse signaling. Little is known about their function in duct formation, whereas key roles in epithelial tissues encompass localization of tight and adherens junctions proteins in the Xenopus epithelial ectoderm[23,24], cell sorting and positioning by local actomyosin contractility at the Xenopus notochord-presomitic mesoderm interface[25], integrin clustering and extracellular matrix assembly in zebrafish somite boundary morphogenesis[26]. In the context of inter-rhombomeric boundaries Ephs and Ephrins inhibit cell intermingling and maintain boundary sharpness by promoting actomyosin cable formation[27,28]. Co-expression of multiple Ephs and Ephrins can confer signaling strength to activate cell detachment during repulsion[29]. We previously showed that EphrinB1 and EphB3b control directional liver progenitor migration into the liver bud by a repulsion-based mechanism during early liver development in zebrafish[30]. Embryos with impaired EphrinB1 or EphB3b function exhibit dysmorphic HPD systems at later stages, suggesting a hitherto unknown function for EphB/EphrinB signaling in HPD morphogenesis.

Here, we present a high-resolution analysis of HPD tube morphogenesis in zebrafish. We show that a single lumen arises by de novo lumen formation from a solid cord-like primordium of unpolarized cells by a cord hollowing mechanism. This multi-step process is driven by dynamic cell rearrangements, which include cell intercalation, promoted by non-muscle myosin II activity. Using genetic approaches, we provide evidence that these processes are regulated by a morphogenetic EphB/EphrinB code of multiple EphrinB ligands and EphB receptors controlling duct differentiation and GB formation in a region-specific fashion.

## Results

**The HPD system forms by de novo lumen formation.** To elucidate the morphogenetic process of HPD formation we first investigated whether the ducts arise from an existing epithelial sheet by a budding or wrapping mechanism, or from unpolarized cells by de novo lumen formation[31]. Overall HPD morphology was visualized by transgenic expression of pan-endodermal XlaEef1a1:GFP[s854], ductal keratin18:GFP (specific for hepatic domains) or immunolabeling of Anxa4, recognized by the 2F11 antibody[17]. Cell polarity and lumen formation were assessed by staining for atypical protein kinase C λ (aPKC) a member of the PAR protein/aPKC complex that localizes to the apical membrane of epithelial cells, or for the tight junction protein ZO-1. Zebrafish larvae were analyzed between 46 hours post fertilization (hpf), when the HPD primordium emerges, and 5 days post fertilization (dpf), when larvae start to feed (Fig. 1a–f). All HPD compartments were assigned based on morphological landmarks. The first signs of tube formation are ZO-1 aggregates, which appear randomly distributed along the central axis of the emerging HPD primordium at 46 hpf (Fig. 1a), indicating that HPD progenitors start to polarize. As early as 48 hpf, the CBD and EPD are distinguishable as the first domains of the HPD, at the proximal end of the liver and pancreas, respectively (Fig. 1b). Multiple, short and disconnected aPKC-positive apical structures within both domains indicate the formation of immature luminal pockets, reminiscent of microlumina described in the murine pancreas[32,33]. In parallel, the HPD-to-gut connection is established through nascent HPA apical pockets at multiple contact points. Hence, all HPD ducts form de novo and not by a budding process from the existing foregut epithelium (Fig. 1a, b). At 52 hpf, the GB primordium is detectable by dense group of aPKC foci and its round morphology at the distal end of the CBD (Fig. 1c), which later will connect to the EHD-CBD lumen via the CD. At this stage, aPKC-positive apical pockets in the CBD and the EPD form a semi-continuous nascent lumen, including gaps and loops of parallel apical structures. The HPD is in most cases connected to the gut at two points revealing that the HPA is still immature (Fig. 1c). Between 52 and 60 hpf, duct maturation results in the consolidation of a single continuous lumen connecting all compartments of the HPD, and duct lengthening (Fig. 1d, g, Supplementary Movie 1). Finally, a refined EHD lumen connecting the HPD to the intrahepatic ductal (IHD) network is clearly distinguishable by 72 hpf (Fig. 1e). Subsequently, the CBD and EPD bend as the connected organs grow and their position within the body adjusts (Fig. 1e–g). Lumen inflation of the HPD compartments occurs after 72 hpf and by 5 dpf (Fig. 1h–j, Supplementary Movie 8), likely due to apical secretion of non-bile fluids from the ductal epithelial cells and bile secretion and transport from the liver starting at 5 dpf[34]. Notably, aPKC staining before 5 dpf indicates HPD cell polarization and lumen formation, however not necessarily an open patent lumen.

These findings show that HPD morphogenesis encompasses a multi-step process, starting with scattered junctional aggregates, which transition to form apical pockets and then fuse into immature ductule morphology, including complex loop-shaped apical structures, and finally remodel into a single central lumen (Fig. 8c). This suggests that HPD tubes form from a solid cord-like primordium, which undergoes de novo lumen formation following a cord hollowing process.

**EphrinB1 and EphB3b control HPD morphogenesis.** To elucidate the molecular mechanisms controlling HPD morphogenesis, we focused on EphrinB1 and EphB3b as candidates, because embryos with compromised function of either factor exhibit dysmorphic HPD systems[30] and EphB/EphrinB signaling governs complex morphogenetic processes[21]. For the analyses, we used *ephrinb1^{nim26}* mutants and generated an *ephb3b^{nim27}* genetic

mutant by Crispr/Cas9 genome editing (Supplementary Fig. 1). We analyzed duct morphogenesis in *ephrinb1* and *ephb3b* mutant embryos at key stages of HPD formation by immunostaining for Prox1, which labels hepatic and pancreatic progenitors, HNF4α for differentiating hepatocytes, and aPKC visualizing apical membranes (Fig. 2, Supplementary Fig. 2). Similar to controls, aPKC-positive apical pockets emerge in *ephrinb1* and *ephb3b*

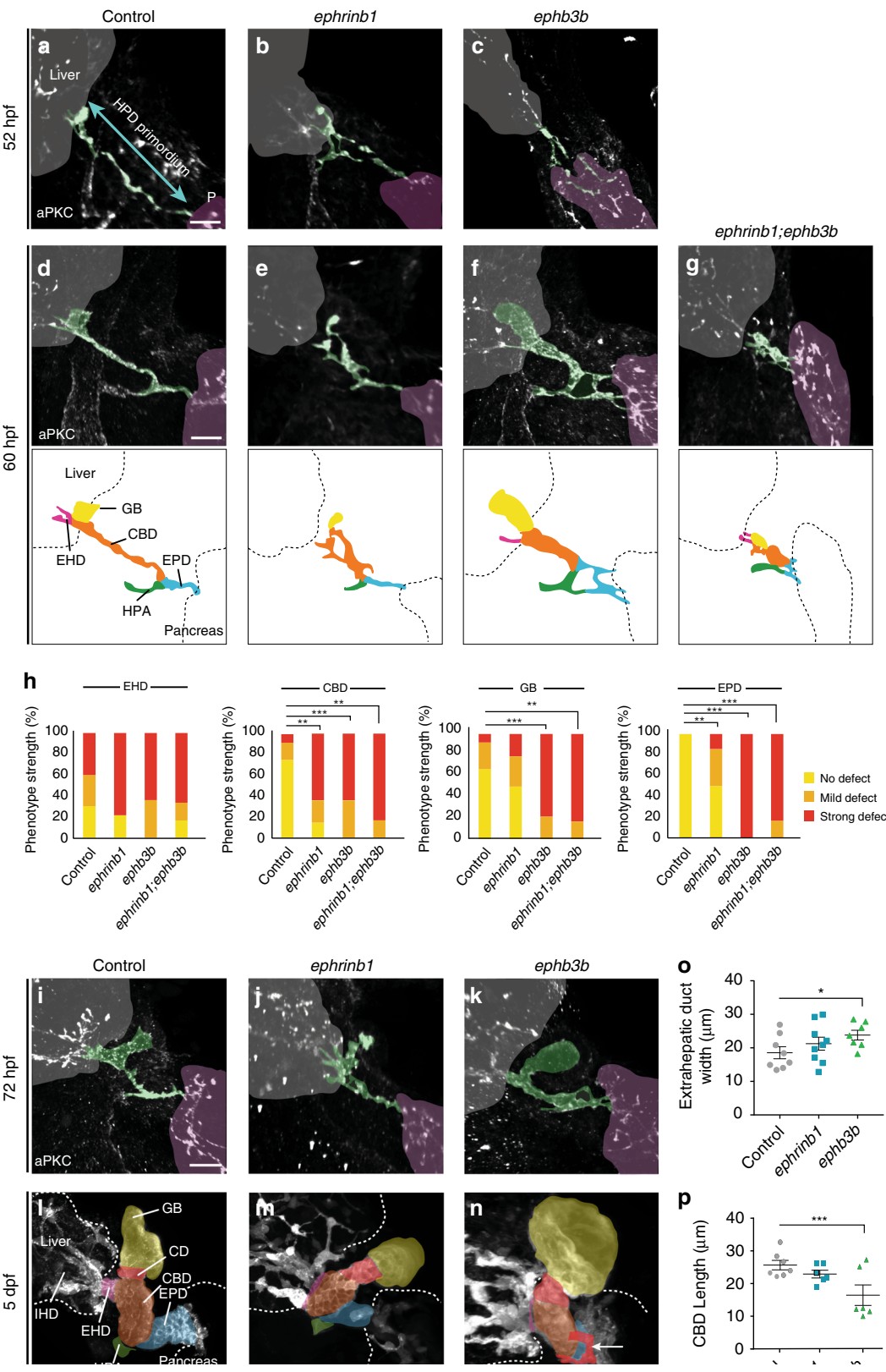

**Fig. 2** EphrinB1 and EphB3b control HPD remodeling in a spatiotemporal fashion. Compared to sibling controls at 52 hpf (**a**), emerging ducts are less mature in *ephrinb1* mutants with more disconnected luminal pockets in EHB and HPA, visualized by apical aPKC (white, $n = 16$, $N = 3$) (**b**). **c** HPD luminal pockets are similarly disrupted in *ephb3b* mutants at 52 hpf, including the EPD ($n = 17$, $N = 2$). **d** Disconnected apical structures resolve into a single immature lumen by 60 hpf in controls ($n = 29$, $N = 2$). **e** Disrupted remodeling causes enlarged aPKC structures and luminal loops in all EHB domains in *ephrinb1* mutants, while the EPD appears normal ($n = 20$, $N = 2$). **f** Resolution of apical structures in *ephb3b* mutants is compromised in all HPD domains, including the EPD ($n = 12$, $N = 1$). **g** All HPD domains exhibit defective tube remodeling in *ephrinb1;ephb3b* mutants ($n = 7$). Schematics (**d–f**) show HPD tube phenotypes at 60 hpf, see below for domain color-code. **h** Quantification of domain-specific HPD tube formation defects in controls ($n = 12$), *ephrinb1* ($n = 14$), *ephb3b* ($n = 14$) and *ephrinb1;ephb3b* ($n = 7$) mutants at 60 hpf. **i–n** Early domain-specific defects persist in *ephrinb1* and *ephb3b* mutants at 72 hpf (**i–k**; controls: $n = 11$, $N = 2$; *ephrinb1*: $n = 5$; *ephb3b*: $n = 12$, $N = 1$) and 5 dpf (**l–n**; control: $n = 10$, $N = 2$; *ephrinb1*: $n = 6$, $N = 1$; *ephb3b*: $n = 10$, $N = 1$). 30% *ephb3b* mutants show in addition ectopic ducts (red; white arrow) bifurcating from the main HPD. Quantification of EHD width (**o**) and CBD length (**p**) in *ephrinb1* and *ephb3b* mutants at 5 dpf. **l–n** Anxa4 (gray) visualizes the HPD system at 5 dpf. HPD domains are color-coded based on morphological landmarks (e.g., HPA): EHD = pink; CD = red; gallbladder = yellow; CBD = orange; EPD = blue; HPA = green. Scale bar = 20 μm; $n$ = sample number, $N$ = number of experiments; Statistical test: **h** = Fisher Exact, **o**, **p** = Student's $t$-test. Errors bar show SEM; *$p < 0.05$, ***$p < 0.001$. Liver and pancreas stainings for **a–f** and **i–k** are shown in Supplementary Fig. 2. Supplementary movies 1–4 and 8–10 show HPD phenotypes at 60 hpf and 5 dpf. Source data are provided as a Source Data file

| Table 1 Domain-specific classification of HPD tube formation defects | | |
|---|---|---|
| **HPD domain** | **Score 1 = mild defect**<br>**(ductal domains show only one of below defects)** | **Score 2 = severe defect (ductal domains show**<br>**more than one of below defects)** |
| Extrahepatic duct (EHD) | Two branches, thinner, enlarged or gap | More than two branches, loops |
| Gallbladder (GB) | A bit longer/more inflated, gap, more than one branch attached distally or enlarged cystic duct | Much longerinflated |
| Common bile duct (CBD) | Long spike, loop, gap or more inflated | Cystic, loops |
| Extrapancreatic duct (EPD) | Long spike, loop, gap or more inflated | Cystic, loops |

mutants at 48 hpf, suggesting that HPD progenitors form a HPD primordium, polarize and initiate the formation of nascent tubes. However, these structures appear slightly disorganized (Supplementary Fig. 2a–c). By 52 hpf, the aPKC structures in the prospective CBD and CBD-EPD-HPA junction are often associated with more disruptions or loops in most *ephrinb1* mutants compared to wild-type (Fig. 2a, b). In contrast to wild-type and *ephrinb1* mutants, *ephb3b* mutants show aPKC loops also in the EPD (Fig. 2a–c). The resolution of these immature lumina into a single continuous lumen primordium in all HPD regions occurs in wild-type by 60 hpf, however, is severely compromised in the absence of EphrinB1 and EphB3b (Fig. 2d–f, Supplementary Movie 1–3). Phenotype variability and penetrance were assessed by quantification of defect severity in the individual HPD domains (Fig. 2h, Table 1, Supplementary Fig. 3). Notably, a single nascent EPD lumen is resolved in most *ephrinb1* mutants (Fig. 2d–e), whereas this is still disorganized and web-like in *ephb3b* embryos (Fig. 2f). *ephrinb1;ephb3b* double mutants exhibit defects in all HPD domains similar to *ephb3b* (Fig. 2g, h, Supplementary Movie 4), disregarding additive or synergistic interactions and corroborating that EphB3b regulates morphogenesis of all HPD domains, while EphrinB1 mediates this process only in a subset of domains. Unresolved loops, enlarged aPKC/luminal structures, an amorph EHD or combinations of these defects, are in both mutants frequently not remodeled by 72 hpf (Fig. 2i–k). Next, we asked whether the observed HPD defects in *ephrinb1* and *ephb3b* mutants persist when digestive activity is starting at around 5 dpf. Anxa4 immunostainings showed dysmorphic ductal compartments in both *ephrinb1* and *ephb3b* mutants, including significantly altered EHD width and CBD length, corresponding with earlier regional remodeling defects (Fig. 2l–p, Supplementary Movies 8–10). In addition, *ephb3b* mutants display larger GB and occasionally ectopic small ducts branching from the EHD, CBD or EPD (30%, Fig. 2n).

Altogether these findings show that EphrinB1 and EphB3b control HPD morphogenesis in a domain-specific manner:

EphrinB1 governs EHB tube morphogenesis, whereas EphB3b is required for the formation of the entire HPD, including the EPD. These only partially overlapping functions imply that HPD morphogenesis may require additional EphrinB ligands and EphB receptors.

**EphrinB1 and EphB3b regulate dynamic CBD cell rearrangements.** Our analysis of lumen formation suggests that the HPD system arises by de novo tube formation. To identify the underlying cell behaviors and morphogenetic mechanism, we first examined the organization of the HPD, focusing on the CBD, by co-staining for apical aPKC and pan-Cadherin, which localizes to lateral membranes in epithelia and is expressed in the HPD endoderm[2]. At 48 hpf, the HPD represents a solid cord consisting of 3–4 cell layers (Fig. 3a), which transforms into a single-layered epithelial duct by 60 hpf (Fig. 3b). Concomitantly, CBD length increases significantly between 52 and 60 hpf in controls, while in both *ephrinb1* and *ephb3b* mutants this extension fails to occur during this time window (Fig. 3e). To distinguish whether CBD tube formation is driven by active cell rearrangement and intercalation, or other mechanisms, such as apoptosis characteristic for cavitation, we analyzed epithelial morphology.

At 60 hpf, when a continuous immature lumen extends throughout the ductal system, epithelial cells are aligned in a single layer with their apical domains forming the arising central lumen (Fig. 3b). Cell shape analyses of individual cells with luminal contribution in wild-type revealed a subset of teardrop-shaped cells during duct remodeling stages (Supplementary Fig. 4a). A teardrop-like morphology is a hallmark of radially intercalating cells[35], suggesting that those cells are actively rearranging within the epithelium. In both *ephrinb1* and *ephb3b* mutants, epithelial organization is disrupted, marked by altered irregular cell shapes and a multi-layered cell organization in areas of immature apical loops (Fig. 3c, d). Ductal cells contributing to ductal loops appear to have a single largely extended or multiple apical domains (Fig. 3c). Cell shape analysis of the CBD revealed

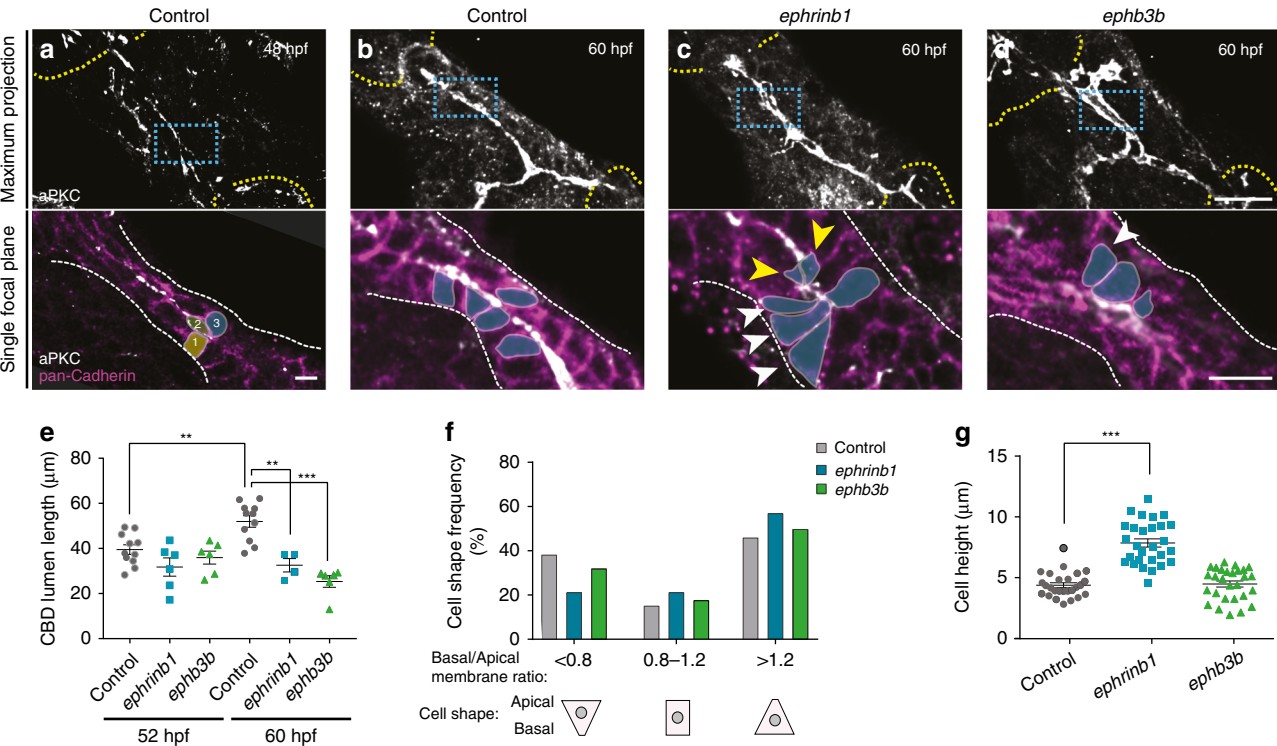

**Fig. 3** Ductal cell rearrangement and lumen resolution require EphrinB1 and EphB3b. **a–d** Overview of the forming HPD epithelium by maximum projection of confocal stacks stained for PanCadherin (magenta) aPKC (white) in control and *ephrinb1* and *ephb3b* mutants. Blue dashed squares indicate magnified CBD containing areas. **a** At 48 hpf, PanCadherin shows multiple cell layers in CBD (e.g., colored cells 1–3). **b** By 60 hpf, the CBD is largely a single-layered epithelium with apical domains facing the central lumen. **c, d** *ephrinb1* and *ephb3b* CBDs show disrupted epithelial organization, including luminal loops, with cells with multiple or extended apical domains (yellow arrowheads), and teardrop shaped-cells with reduced apical domains (white arrowhead). **e** Quantification of the longest continuous aPKC staining as a measure for CBD lumen elongation in controls, *ephrinb1* and *ephb3b* mutants between 52 and 60 hpf. **f** Basal/apical membrane ratio was used to quantify CBD cell shapes in control (*n* = 26 cells), *ephrinb1* (*n* = 28) and *ephb3b* (*n* = 28) embryos at 60 hpf. **g** Quantification of CBD cell height (distance between apical and basal membranes) in control (*n* = 26 cells), *ephrinb1* (*n* = 28) and *ephb3b* (*n* = 28) embryos at 60 hpf. Scale bars: **a** = 5 µm, **b–d** = 20 µm. Statistical test: **e, g** = Student's *t*-test, **f** = Fisher Exact. Errors bar show SEM; **p < 0,01, ***p < 0,001. Source data are provided as a Source Data file

a clear increase of teardrop-shaped cells facing the apical lumen in both *ephrinb1* and *ephb3b* mutants at the expense of those with opposite orientation and more columnar cells (Fig. 3f). In both mutants, these changes are associated with an increase in cell height (Fig. 3g). Impaired cell rearrangement in *ephrinb1* and *ephb3b* mutants results in a failure of CBD extension (Fig. 3e). Finally, active cell rearrangement is supported by sparse labeling of small clones with the membrane-tethered fluorescent protein lyn-Tomato in wild-type, revealing HPD cells extending small filopodia-like protrusions, including several towards the forming lumen (Supplementary Fig. 4b). Given the known role of EphB/ EphrinB signaling in cell proliferation[21], we assessed its contribution to HPD morphogenesis. Neither cell proliferation rate nor overall HPD cell number are significantly changed in *ephrinb1* and *ephb3b* mutants compared to controls (Supplementary Fig. 6), indicating that EphrinB1 and EphB3b control HPD morphogenesis mainly by alternative mechanisms. Together these data indicate that scattered apical pockets remodel and coalesce into a continuous nascent lumen by dynamic cell shape changes and active intercalation-like cell rearrangement, transforming a compact cord-like HPD primordium into a mono-layered epithelial duct. EphrinB1 and EphB3b play a crucial role in regulating these cell shape changes essential for HPD tubulogenesis.

**Non-muscle myosin activity mediates HPD lumen remodeling.** The dynamic changes in epithelial organization of the HPD

during duct differentiation and the related defects in *ephrinb1* and *ephb3b* mutants suggest an active contribution of cytoskeletal remodeling and actomyosin contractility to the tube maturation process[36]. First, we assessed whether myosin II contractility is required for HPD formation by incubating wild-type embryos with the selective pMLC inhibitor blebbistatin during lumen remodeling (Fig. 4a). Blebbistatin incubation consistently caused luminal gaps and loops in the HPD of 60 hpf embryos (Fig. 4b–d), reminiscent of the lumen remodeling defects observed in *ephrinb1* and *ephb3b* mutants. Moreover, remodeling defects are associated with rounded and/or elongated epithelial cells and a significantly elongated CBD (Fig. 4e). These findings show that myosin II contractility is required for HPD remodeling into a mature tube, possibly contributing to multiple steps of cell shape changes leading to the rearrangement of polarizing cells to form a central lumen. To elucidate the function of myosin activity, we determined its subcellular distribution. Staining showed that actin and active non-muscle myosin II, visualized by phosphorylation of its regulatory light chain (pMLC), are enriched apically and at lower levels at lateral and likely at basal membranes during stages of HPD remodeling (Fig. 4f). Myosin contractility is therefore a good candidate to regulate apical domain size in the differentiating HPD. We determined apical pMLC levels focusing on *ephrinb1* mutants, which exhibit distinct cell shape change defects in the CBD at 60 hpf (Fig. 3f, g). Quantification of staining intensity showed an up to six-fold increase of apical pMLC levels in *ephrinb1* mutants compared to

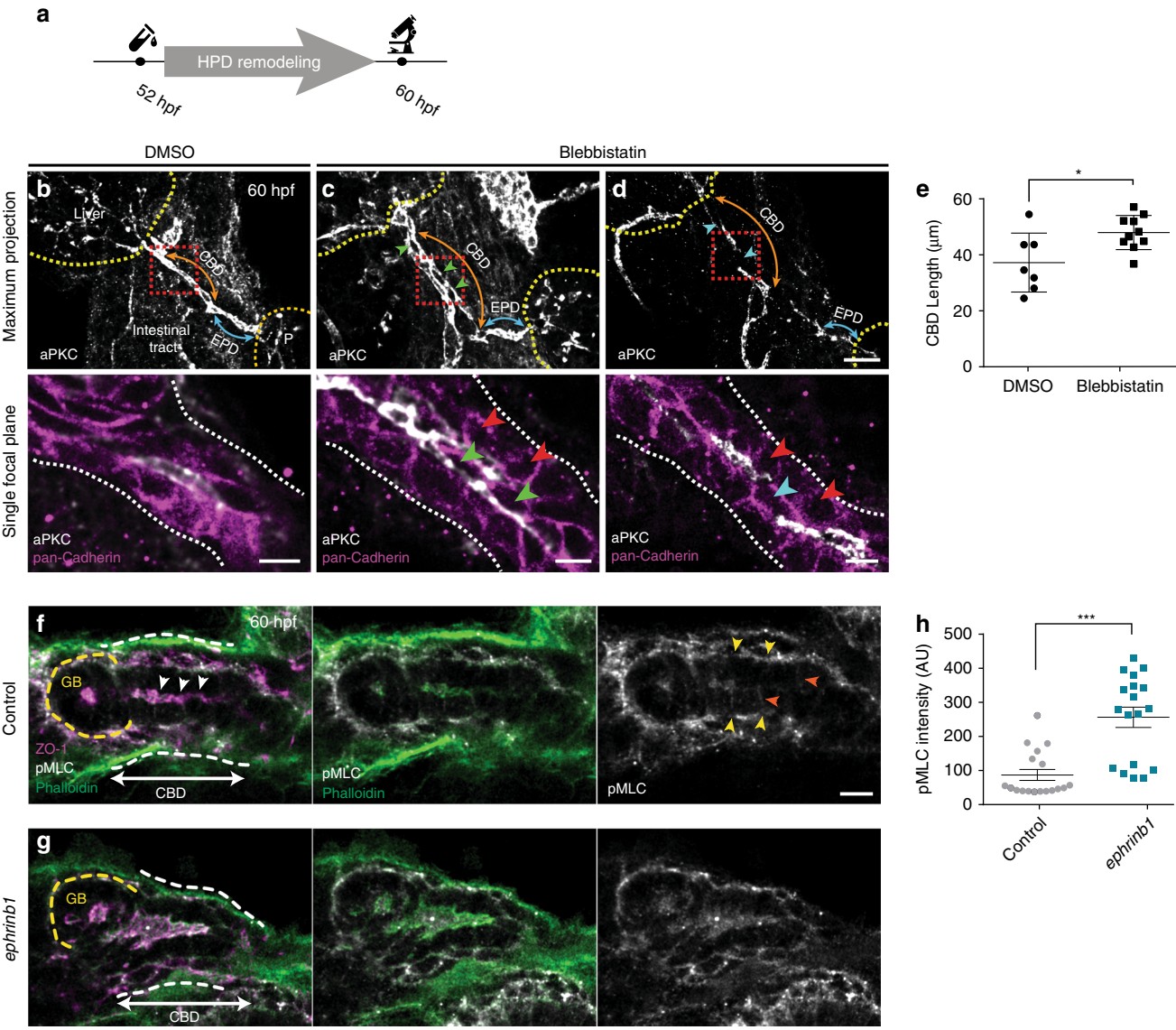

**Fig. 4** HPD differentiation requires EphrinB1-controlled actomyosin contractility. **a** Experimental scheme for 65 μM blebbistatin incubation during HPD remodeling from 52–60 hpf. **b**–**d** Blebbistatin-treated embryos show two types of EHB epithelial defects: **c** extensive luminal loops (45.5%, n = 11, N = 2) and **d** gaps in the emerging apical lumen (45.5%, n = 11, N = 2; and 9% wild-type like). The nascent EPD lumen is mostly resolved, consistent with the treatment window; scale bars = 15 μm. PanCadherin staining (magenta) of magnified views of CBD (red dashed box) shows rounder epithelial cells (red arrowheads), a multi-layered configuration with loops and cells exhibiting extended apical domains (green arrowheads) or gaps and cells with multiple apical domains (blue arrowhead) in treated embryos compared to controls; scale bars = 5 μm. **e** Quantification of CBD length shows significant increase upon blebbistatin treatment compared to controls. **f**, **g** Actin, pMLC and ZO-1 localize apically (white arrowheads) in the forming CBD; whether pMLC is at the apical membrane and/or the junctional belt cannot be distinguished. pMLC is at the duct mesoderm border (yellow arrowheads) and lower levels at the lateral membrane (orange arrowheads) 60 hpf; scale bars = 5 μm. **h** Quantification of apical pMLC expression in CBD cells of 60 hpf control (n = 3) and ephrinb1 embryos (n = 3). For quantification strategy see Supplementary Fig. 5. Statistical test: **e**, **h** = Student's t-test. Error bars show SEM; *p < 0,05, ***p < 0,001. Source data are provided as a Source Data file

controls, suggesting hyper-contracted apical domains contribute to the altered cell shapes in the CBD (Fig. 4f–h, Supplementary Fig. 5). These data demonstrate that EphrinB1 contributes to HPD cell rearrangement and lumen resolution by directly or indirectly regulating apical pMLC levels and distribution.

**EphB/EphrinB signaling is required for HPD lumen remodeling.** To distinguish whether HPD malformations in ephrinb1 and ephb3b mutants are the result of a requirement for EphrinB1 and EphB3b in HPD remodeling or secondary to earlier progenitor positioning defects[30], we blocked EphB/EphrinB

signaling specifically during the time window of duct remodeling. For this we expressed EphrinB1[EC], the extracellular domain of EphrinB1 which acts as a secreted, dominant-negative form[37]. For conditional ubiquitous expression transgenic UAS:ephrinB1[EC] fish were crossed with hsp70l:Gal4 fish and the embryos exposed to heat-shock at key stages of HPD remodeling and assessed at 60 hpf when a single HPD lumen is established (Fig. 5a, b). The majority of manipulated embryos display severe morphological HPD defects upon blocking EphB/EphrinB signaling around the onset of remodeling at 50 hpf or when remodeling is ongoing at 54 hpf. Defects include luminal loops, dysmorphic and enlarged immature luminal domains

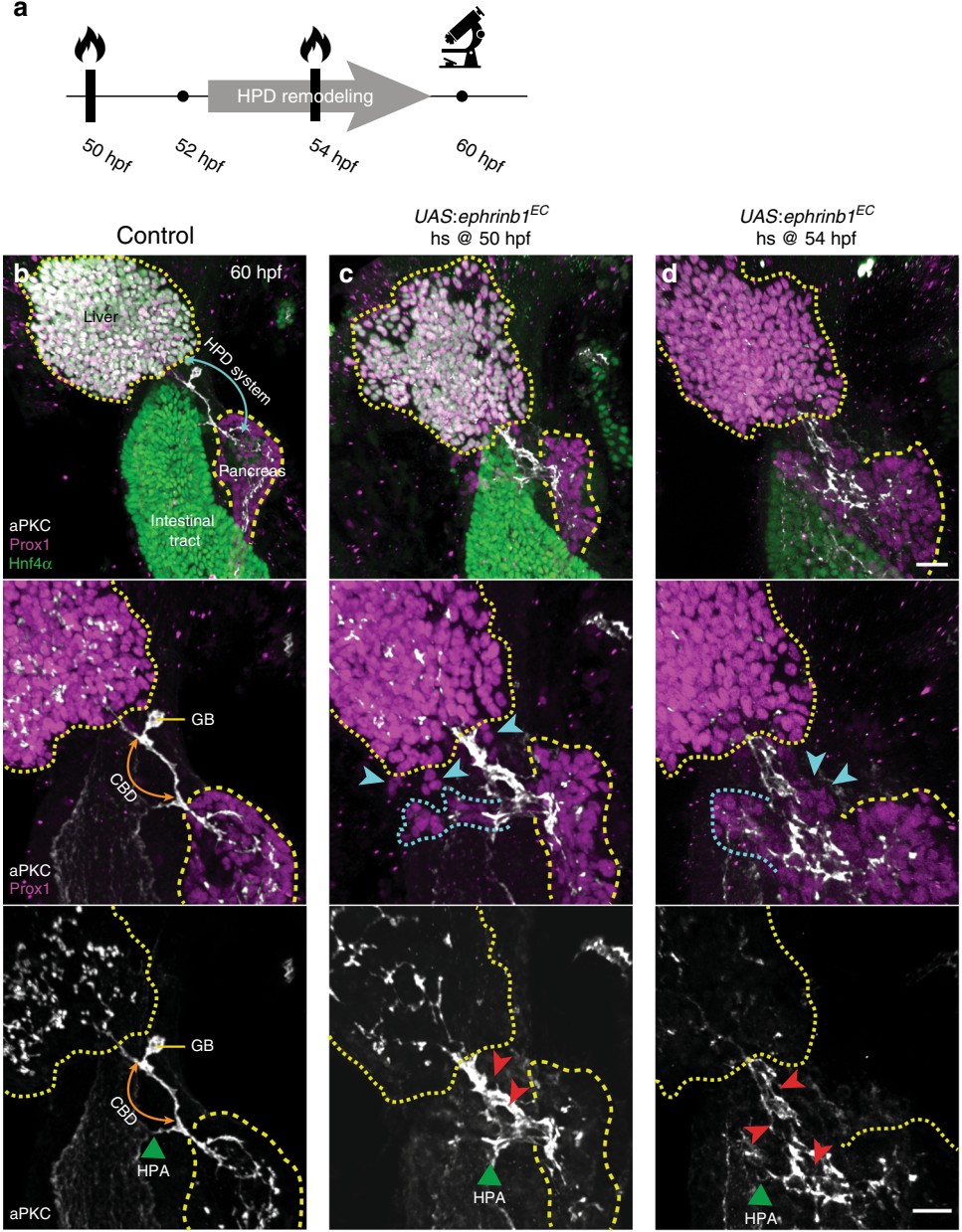

**Fig. 5** EphB/EphrinB signaling controls HPD lumen morphogenesis independently of other morphogenetic roles. **a** Conditional *ephrinb1EC* expression is induced by heat-shock prior to or during CBD remodeling, at 50 hpf or 54 hpf, respectively. Embryos were analysed at 60 hpf (heat-shock at 50 hpf: $n = 7$, heat-shock at 54 hpf: $n = 7$, both $N = 1$). **b**–**d** Hnf4α (green), Prox1 (magenta) and aPKC (gray) labeling show that unlike in controls (**b**), conditional *ephrinb1EC* expression at 50 hpf (**c**) and 54 hpf (**d**) leads to HPD, liver and pancreas morphology defects, ectopic Prox1-positive cells throughout the HPD (blue arrowheads), with a subset associated with ectopic lumina bifurcating from the main duct (outlined with blue dashed line) disrupted and disorganized HPD lumen (red arrowheads, **c**, **d**). Yellow dotted lines: liver, yellow dashed line: pancreas; blue arrow: HPD system, orange arrow: CBD, green arrowhead: HPA

throughout the HPD and a severely hypoplastic or absent GB (Fig. 5c, d). In addition, in most embryos the morphology of the pancreas is dysmorphic, with some embryos displaying ectopic groups of Prox1+/Hnf4α− cells that frequently appear connected by aPKC+ ductal structures to the EPD or pancreas (Fig. 5c, d), suggesting impaired integrity of the pancreatic epithelium and its expansion into ectopic locations. Hence, EphB/EphrinB signaling is required for HPD duct remodeling and EphrinB1 and EphB3b could play multiple independent and sequential roles in liver, pancreas and HPD development. Moreover, phenotypes upon blocking EphB/EphrinB signaling are often more severe than those observed in *ephb3b* and in particular *ephrinb1* mutants

alone, supporting that additional EphrinBs and EphBs may be involved in HPD development.

**EphrinB2a and EphB4a regulate GB and CBD formation.** In light of the only partial overlap of *ephrinb1* and *ephb3b* mutant HPD phenotypes and the more severe malformations observed upon blocking EphB/EphrinB signaling, we assessed the involvement of additional EphrinB ligands and EphB receptors in HPD development. First, we determined the spatial expression of EphrinB1 and EphB3b by immunostaining. During the onset of HPD tube formation, at 48 hpf, EphrinB1 is expressed in the liver

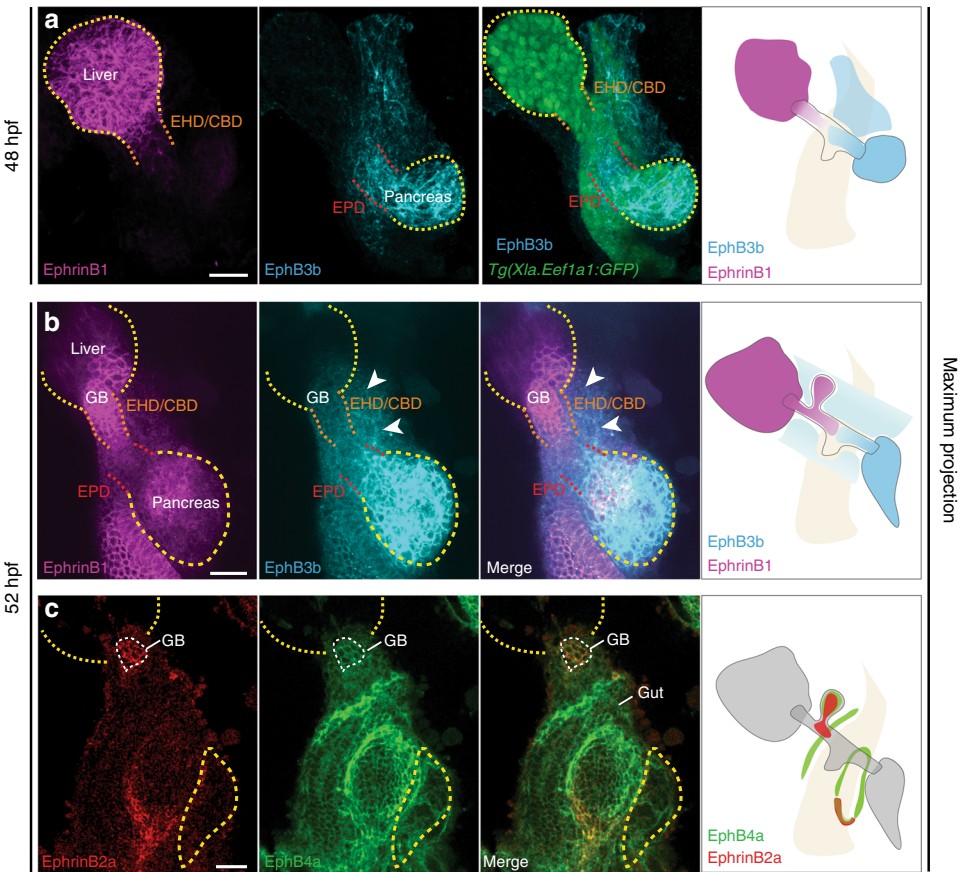

**Fig. 6** Regionalized HPD endoderm and mesoderm expression of multiple EphrinBs and EphBs. **a** At 48 hpf, EphrinB1 is strongly expressed in the liver and at lower levels in the HPD primordium close to the liver. EphB3b expression is high in the pancreas and in the HPD primordium close to the pancreas and lower close to the liver, as well as very low in the HPD-adjacent mesoderm; pan-endodermal *tg(Xla.Eef1a1:GFP)* highlights the digestive system. **b** At 52 hpf, EphrinB1 is expressed in the liver (yellow dotted line) and EHB (white arrow), while EphB3b is highly expressed in the pancreas (yellow dotted line) and at low levels in the EPD (white arrow) and HPD mesenchyme (white arrowheads). **c** EphrinB2a and EphB4a are co-expressed in the gallbladder (white dashed line). EphrinB2a is present in the posterior region of the swim bladder. EphB4a is in the mesenchyme along the gut tube and around the swim bladder. Scale bar = 30 µm

and at lower levels in the HPD primordium next to the liver, whereas EphB3b shows strong expression in the pancreas and HPD adjacent to the pancreas and weak expression adjacent to the liver and HPD-surrounding mesenchyme (Fig. 6a). At 52 hpf and as the HPD starts differentiating, EphrinB1 is expressed throughout the liver and in the EHD, GB, CBD and intestinal epithelium, while EphB3b is expressed highly in the pancreas and at lower levels in the EPD and mesoderm surrounding the HPD system (Fig. 6b). Given their bidirectional signaling capabilities, this expression profile suggests that EphrinB1 may interact with EphB3 at (i) the endoderm-mesoderm interface for the EHD, GB and CBD and (ii) within the duct at the CBD-EPD boundary (Fig. 6b). Notably, EphrinB1 expression in *ephb3b* mutants expands from the CBD into the EPD (Supplementary Fig. 7), possibly because of the absence of EphB3b-mediated membrane removal by endocytosis[38]. Alternatively, EPD cells express EphrinB1 due to the release of repressive EphB3b effectors at the transcriptional level.

Next, we investigated EphrinB2a and EphB4a as additional regulators of HPD formation, since they frequently act in concert with EphrinB1 and/or EphB3 in complex morphogenetic processes[21], such as vascular morphogenesis[39], cell positioning in the intestine[40] or tissue separation[27,29]. To assess their spatiotemporal expression, we generated polyclonal antibodies revealing EphrinB2a expression in the GB primordium and at lower levels in adjacent parts of the CBD at 52 hpf, while EphB4a

is co-expressed in the GB and non-HPD mesoderm (Fig. 6c). Hence, these four genes are expressed in a regionalized and partly overlapping pattern during HPD development and in differentiated ducts. In addition, mammalian conservation was suggested by expression analysis of human tissue samples, revealing *EPHRINB1, EPHRINB2*, and *EPHB4* expression in large biliary ducts and *EPHRINB1, EPHRINB2, EPHB3*, and *EPHB4* in extrahepatic cholangiocarcinoma tissue, serving as closest avatar, since access to healthy tissue is difficult (Supplementary Fig. 8).

To assess EphrinB2a and EphB4a functions in HPD development, we analyzed mutants for HPD lumen formation by aPKC staining at 60 hpf and HPD morphology by Anxa4 staining at 5 dpf. *ephrinb2a* mutants exhibit unresolved loop-like lumina, both in the GB and CBD, at 60 hpf (Fig. 7c, f, Supplementary Movie 5), and substantially smaller GBs and dysmorphic CD and CBD at 5 dpf (Fig. 7g, i, Supplementary Movie 11). Given the EphrinB1 and EphrinB2a co-expression in overlapping HPD domains, we assessed their genetic relationship in double mutants. Most *ephrinb1;ephrinb2a* HPD phenotypes are more severe, including cysts in the EHD, GB and CBD region at 60 hpf, consistent with additive phenotypes (Fig. 7b–d, f, Supplementary Movie 6). Similarly at 5 dpf, *ephrinb1; ephrinb2a* double mutants display significantly smaller GBs and short dysmorphic EHDs and CBDs (Fig. 7h-j, l, m, Supplementary Movie 12). *ephb4a* mutants showed a transiently smaller GB primordium compared

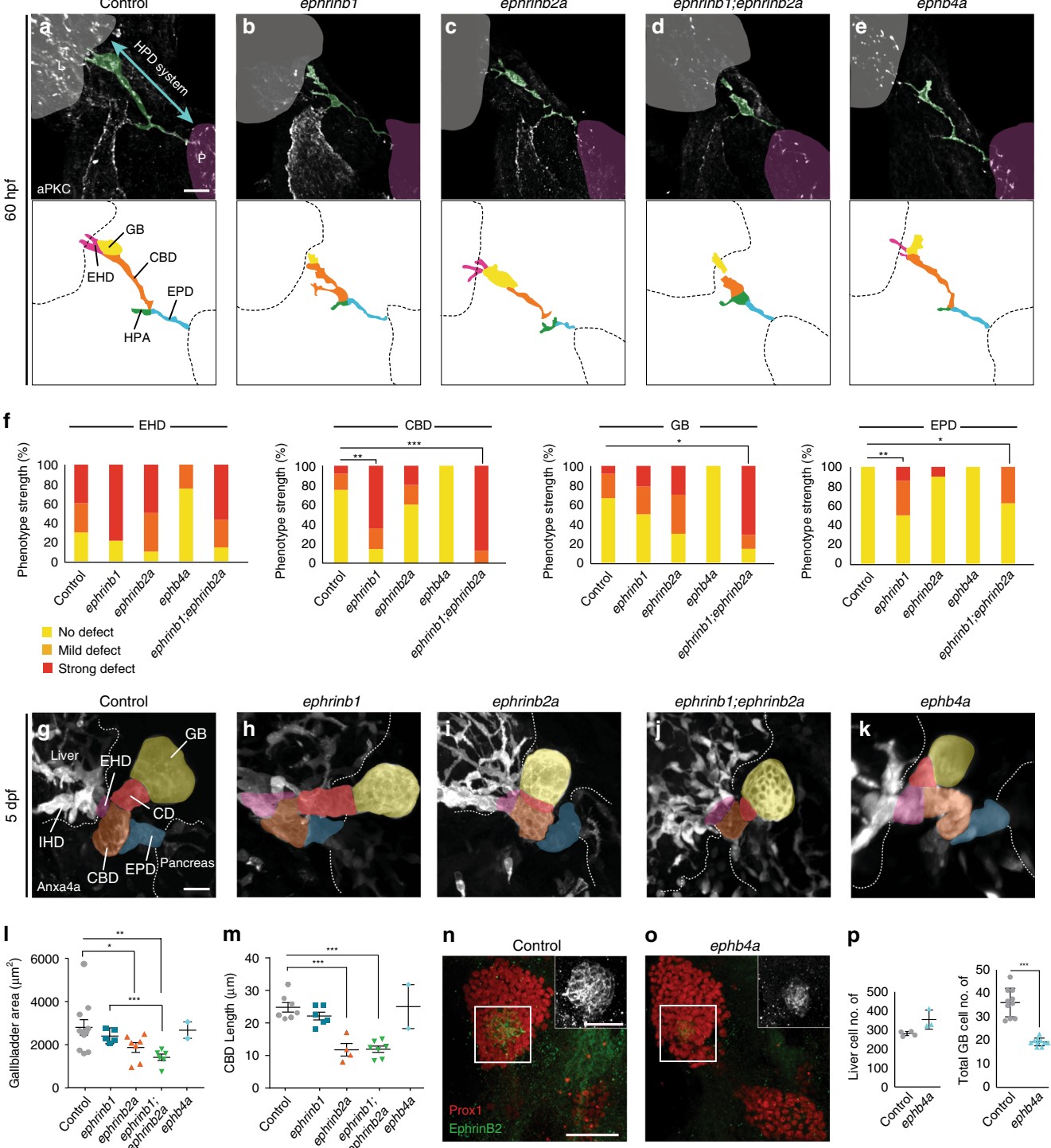

**Fig. 7** EphrinB2a and EphB4 functions in gallbladder and CBD formation. Unlike sibling controls (**a**), *ephrinb1* mutants (**b**) show disrupted ducts in the EHB region. *ephrinb2a* mutants (**c**) exhibit dysmorphic gallbladder and CBD (46.1% show the full phenotype, 53.9% show milder phenotype, *n* = 13, *N* = 2). **d** *ephrinb1;ephrinb2a* double mutants show additive EHB duct defect (100% *n* = 8, *N* = 1). **e** *ephb4a* mutants show no apparent HPD defect (*n* = 4, *N* = 1). Schematics of immature HPD lumina show in **a-e** in domain-specific color code. **f** Quantification of *ephrinb1*, *ephrinb2a*, *ephb4a* and *ephrinb1; ephrinb2a* double mutant HPD formation defects at 60 hpf, scored for individual HPD domains (control: *n* = 12; *ephrinb1*: *n* = 14; *ephrinb2a*: *n* = 10; *ephb4a*: *n* = 4, *ephrinb1;ephrinb2a*: *n* = 8); see Supplementary Fig. 3 for averaged scores from two assessors. **g–k** Early duct morphogenesis defects cause dysmorphic HPD appearance at 5 dpf in *ephrinb1*, *ephrinb2a* and *ephrinb1;ephrinb2a* double mutants, while the HPD in *ephb4a* mutants appears control-like. (control: *n* = 10, *N* = 2; *ephrinb1*: *n* = 6, *N* = 1; *ephrinb2a*: *n* = 7; *ephrinb1;ephrinb2a*: *n* = 6; *ephb4a*: *n* = 2) **l** Area quantification shows the gallbladder is significantly smaller in *ephrinb1; ephrinb2a* double mutants at 5 dpf. **m** Quantification of CBD length at 5 dpf shows significant differences in *ephrinb2a* and *ephrinb1;ephrinb2a* mutants. **n–p** In contrast to controls (**n**), fewer EphrinB2a-positive gallbladder precursors form in *ephb4a* mutants at 52 hpf (**o**). **p** Quantification of liver and gallbladder progenitor numbers (control and *ephb4a n* = 10). **a-e**: Scale bar = 15 μm, **g–k** = 20 μm. Statistical test: **f** = Fisher Exact, **l**, **m**, **p** = Student's *t*-test. Error bars show SEM; *\*p* < 0,05; *\*\*p* < 0,01; *\*\*\*p* < 0,001. Supplementary movies 1, 2, 5–9, 11–13 show HPD phenotypes at 60 hpf and 5 dpf. Source data are provided as a Source Data file

to unchanged liver progenitor numbers at 52 hpf (Fig. 7n–q), but no detectable change in the HPD at 60 hpf and 5 dpf (Fig. 7e, f, k–m, Supplementary Movies 7, 13), indicating compensation by other factors. Likewise, *ephb3b; ephb4a* double mutants showed no additional phenotypes to those of *ephb3b* mutants (data not shown). Together these data demonstrate compartment-specific expression and requirements for EphrinB1, EphrinB2a, EphB3b, and EphB4a in HPD duct remodeling and GB formation, revealing an EphB/EphrinB code-like control of HPD morphogenesis, including signaling interactions at multiple endoderm-mesoderm and intra-endodermal tissue interfaces.

## Discussion

In this study, we show that the HPD system differentiates by a dynamic morphogenetic process, in which a mature tube forms by a multi-step cord-hollowing mechanism. This includes remodeling from multiple nascent lumina into a single lumen, which requires active cell intercalation and actomyosin contractility. We identify EphB/EphrinB signaling as a key regulator of this step. Specifically, multiple EphB receptors and EphrinB ligands are required for HPD morphogenesis in a compartment-specific fashion. In light of the regionalized and partly overlapping ligand and receptor expression in the HPD endoderm and surrounding mesoderm we propose that compartment-specific duct remodeling is controlled by a morphogenetic EphB/EphrinB code that involves signaling at multiple tissue interfaces (Fig. 8a, b).

The HPD system primordium forms between the outgrowing liver and ventral pancreas buds[2,15,41] and we show that HPD tubes form concomitant to the differentiating IHDs and IPDs. Polarity acquisition of HPD progenitors is marked by the emergence of multiple tight junction protein ZO-1 aggregates, which subsequently mature into nascent apical aPKC$^+$ microlumina. Upon expansion, they connect with adjacent microlumina coalescing into a continuous nascent lumen, transitioning through intermediates of luminal loops and occasionally parallel lumina. Our findings thus demonstrate that HPD de novo lumen formation occurs by a dynamic multi-step process to generate epithelial tubes by cord hollowing (Fig. 8c), similar to lumen morphogenesis in the zebrafish gut[42–44] and the steps underlying ductal plexus formation in the mouse pancreas[32,33,45].

Our high-resolution analysis further revealed that cell intercalation is part of the cell rearrangements transforming the multi-layered cord into a single-layered ductal epithelium. HPD teardrop-shaped cells extend apical filopodia during lumen remodeling similar to radial intercalation of multi-ciliated cells into the epidermis in *Xenopus*[46] and extending mammary gland ductules[35] corroborating the role of cell intercalation in driving epithelial tube elongation, such as *Drosophila* trachea or in the vertebrate kidney[47,48], whereby the exact mode (e.g., radial, rosette mediated) may differ.

Adding to the mechanistic understanding of cord hollowing, we demonstrate the importance of actomyosin contractility. Given the diverse roles of myosin II in morphogenesis, it is likely involved in multiple steps of tubulogenesis. First, apical contraction by actomyosin activity controls lumen size[49,50], while inhibition leads to cell spreading. Controlled apical actomyosin contractility may allow just enough cell spreading to ensure fusion of nascent microlumina, while loss of contractility leads to excessive lumen expansion. In line with this, HPD progenitors exhibit high apical pMLC and blocking myosin II during CBD lumen resolution results in an expanded luminal network. Conversely, an increase of apical pMLC levels in *ephrinb1* mutant CBDs is associated with an increase of teardrop–shaped cells and domain shortening. This suggests that EphrinB1 promotes the

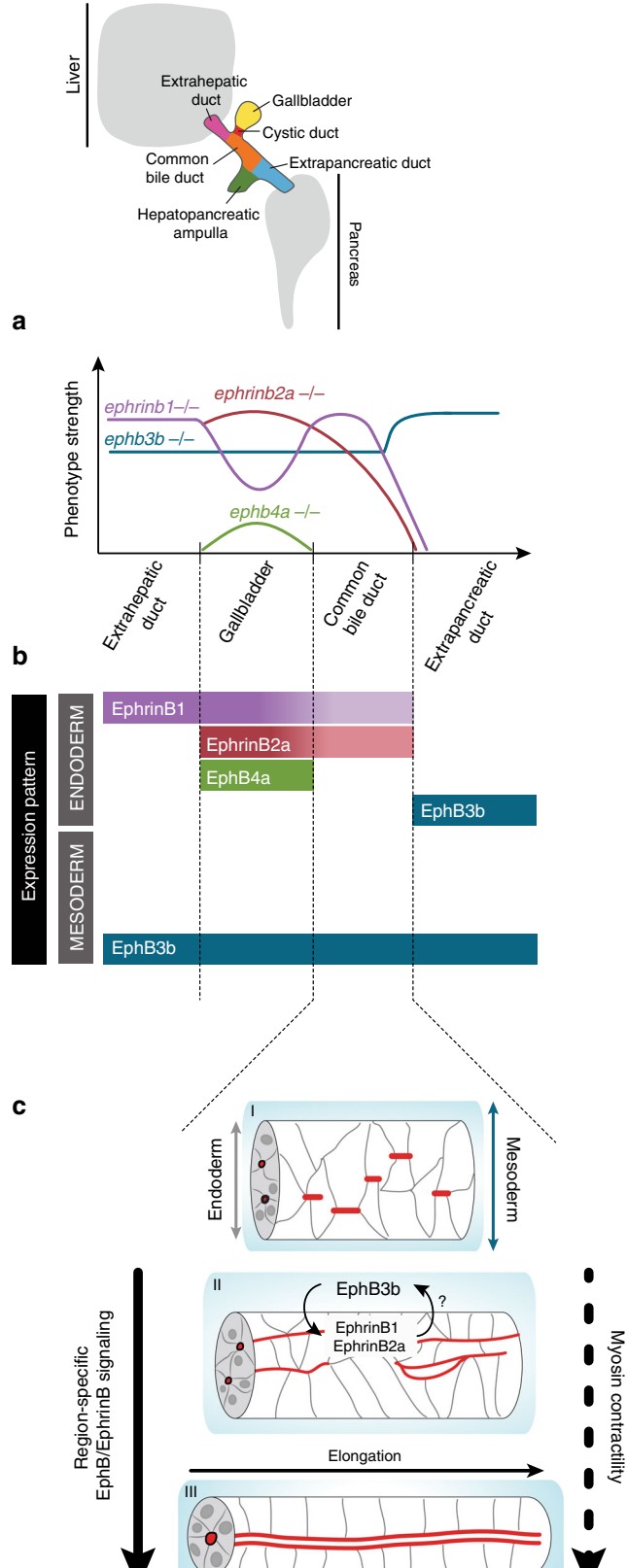

acquisition of distinct apical myosin II levels, which prevent cell spreading and thereby ensure moderate contractility critical for coordinated cell intercalation and balanced apical expansion during CBD formation. We thus propose that tight spatial and temporal control of apical membrane size is essential for lumen coalescence in cord hollowing. Constraining lumen size by

**Fig. 8** An EphB/EphrinB code and myosin II contractility control HPD tubulogenesis. **a** Graphic representation of domain-specific phenotypes observed in *ephrinb1*, *ephrinb2a*, *ephb3b* and *ephb4a* mutants; y-axis indicates overall severity of phenotypes. **b** Graphic representation of domain-specific expression of EphrinB1, EphrinB2a, EphB3b and EphB4a in HPD endoderm and surrounding mesoderm at 52 hpf. **c** Working model of HPD de novo lumen formation focused on the CBD domain. Lumen formation occurs by a multi-step cord hollowing mechanism, which starts with scattered junctional aggregates, transitions through apical/luminal pocket formation and their subsequent coalescence and finally resolves into a single lumen. Apical/luminal pocket coalescence is promoted by ductal EphrinB1/EphrinB2a interaction with mesodermal EphB3b at the tissue interface and actomyosin contractility

myosin II is important during ascidian notochord formation[51], lumen expansion in zebrafish and murine blood vessel development[49,50] and branching of pancreatic ducts[52]. Furthermore, blocking myosin II and its localization to actin fibers at the leading edge of migrating cells[53] would compromise cell intercalation and hence formation of a single-layered epithelial duct. Finally, low pMLC at the lateral side of CBD cells indicates that it may control membrane shortening, similar to ascidian endoderm invagination[54]. Altogether, our data show a mechanistic link between EphrinB1 and myosin II pathway. EphrinB1 could directly control apical pMLC localization, or indirectly, for instance via RhoA and ROCK, known effectors of EphB/EphrinB signaling[20]. Co-expression of EphrinB1 and EphrinB2a combined with additive CBD defects in double mutants point to similar functions for EphrinB2a in this process. Milder cell shape defects in *ephb3b* CBDs suggest that EphB3b controls tubulogenesis only partly by EphrinB-mediated apical pMLC levels. EphB/EphrinB signaling could further contribute to other roles in cell migration/intercalation and/or basolateral membrane shortening during HPD tube formation. The importance of cytoskeletal remodeling in biliary ducts is highlighted by ductal defects in zebrafish mutants for Hippo-Yap pathway component Nf2 and the serine-threonine kinase Cdk5[55,56].

Repulsion-based activity of regionalized Ephs and Ephrins is known to coordinate complex morphogenetic developmental processes, such as rhombomere boundary formation in the hindbrain[27] or axon guidance to generate topographic maps in the retinotectal system[21,57]. During early organogenesis, EphBs and EphrinBs direct the asymmetric migration of hepatoblasts to their final position[30]. Yet for the HPD system, we show that regionalized EphB and EphrinB activity is required for compartment-specific tube morphogenesis (Fig. 8a). Both the spatiotemporal expression and mutant phenotypes suggest that EphB/EphrinB signaling regulates HPD morphogenesis from the onset, and early progenitor positioning may partly contribute. This uncovers a putatively more general role of Eph/Ephrin signaling in directing tube morphogenesis in complex ductal systems, because regionalized expression is reported in the mouse kidney[58]. Specifically, EphB3b is required for the remodeling of microlumina into a single lumen throughout the HPD, while EphrinB1 controls only the remodeling of the EHB, but not the EPD. EphB3b therefore interacts with an additional currently unknown EphrinB ligand in EPD morphogenesis, or acts in a ligand-independent fashion[59]. EphrinB2a and EphB4a regulate GB size, either by mediating progenitor specification, aggregation of GB progenitors into the prospective GB primordium or by proliferation. Therefore, regionalized members of the same gene family ensure efficient and coordinated morphogenesis of all HPD parts, resulting in compartment-specific morphology of the individual ducts. First insights of how these complex expression

patterns arise, come from analysis of *sox9b* mutants exhibiting a mis-patterned and dysmorphic HPD system[15,16]. We show that EphrinB1 expression in the CBD is reduced in 60% of *sox9b* mutants (data not shown), indicating that Sox9b controls HPD patterning and morphogenesis via independent effectors.

Given the bidirectional signaling capabilities of EphB/EphrinB signaling, the regionalized expression of two EphrinB ligands and two EphB receptors during HPD development allows different signaling possibilities. EphrinB1/EphrinB2a expression in all EHB compartments, and EphB3b expression in the EHD surrounding mesoderm and the EPD endoderm (Fig. 8b), suggests signaling in *trans* across two receptor-ligand interfaces: (i) remodeling of the EHB ducts can be mediated by signaling across the endoderm-mesoderm boundary, with EphrinB1/EphrinB2a reverse signaling controlling the dynamic cell rearrangements in the ductal epithelium. (ii) EphrinB1/EphB3b interactions at the CBD-EPD boundary may regulate duct remodeling via reverse signaling in the EphrinB1-positive CBD, and by forward signaling in the EphB3b bearing EPD. This disruption of the CBD-EPD junction is similar to pancreaticobiliary maljunctions of patients with two-way reflux of bile and pancreatic juice, inflammation and metaplasia of ductal cells[60]. Notably, ductal tight junction proteins and myosin light chain kinase are altered in these cases, corroborating the myosin II requirement for HPD formation shown in our study. The additive EHB defects in *ephrinb1;ephrinb2a* double mutants indicate that both ligands interact with mesodermal EphB3b, reflecting increased signaling strength, similar to what has been observed in *Xenopus* germ layer separation[29] and in line with promiscuous ligand-receptor interactions[20].

Blocking EphB/EphrinB signaling by dominant negative EphrinB1^EC expression, demonstrates that HPD lumen resolution is specific and independent from earlier EphB/EphrinB functions in left-right signaling or asymmetric liver progenitor positioning[30,61], revealing temporally distinct sequential roles in organogenesis. Finally, the expression of EPHBs and EPHRINBs in human indicates a conserved role in epithelial maintenance of large biliary ducts and possible HPD morphogenesis across vertebrates.

This systematic analysis describes how functional HPD tubes form to connect the liver and pancreas to the intestine, establishing a reference for future studies of HPD development and to further our understanding of congenital cases of HPD malformation. The identification of regionalized activity of multiple EphrinB ligands and EphB receptors uncovers a paradigm for coordinating the morphogenesis of multi-compartment ductal systems and subsequent maintenance of tissue integrity.

## Methods

**Zebrafish stocks**. Adult zebrafish and embryos *(Danio rerio)* were kept and raised according to standard laboratory conditions[62]. All experiments were performed according to ethical guidelines approved by the Danish Animal Experiments Inspectorate (Dyreforsøgstilsynet) or the UK Home Office under the Animals (Scientific Procedures) Act 1986. The following strains were used: *Tg (UAS: ephrinb1^EC)^nim25* [30], *Tg(hsp70l:Gal4)^fci1* (gift from David Wilkinson), *ephrinb1^nim26* [30], *ephrinb2a^hu3393* and *ephb4a^hu3378* (Stemple Laboratory, direct submission to ZFIN), *Tg(keratin18:GFP)^p314* [63] and *Tg(Xla.Eef1a1:GFP)^s854* [41].

**Generation of genetic mutants**. The *ephb3b^nim27* mutant allele was generated by CRISPR/Cas9 injections into one-cell-stage zebrafish embryos[64]. A single sgRNA targeting the sequence 5′-GGG CCA TGA CTG AGC TGG CC-3″ in the second exon of EphB3b (Supplementary Fig. 1) was created by annealing and cloning two complementary oligonucleotides into the pDR274 backbone (Addgene #42250). In vitro synthesis of the sgRNA was performed using the T7 RiboMAX™ Large Scale RNA Production System (Promega). Mutations were analyzed by amplicon restriction using the primers *ephb3b*F 5′-GGCCTGTTGTTTCCTCAGACT-3′ and *ephb3b*R 5′-GCAAAACATGACCTTGTTGA-3′, followed by XcmI restriction and verified by sequencing. A carrier fish transmitting a 4 bp deletion affecting the ligand-binding domain was used to generate a stable line.

**Table 2 PCR and restriction digest information for mutant genotyping**

| Mutant | Forward primer | Reverse primer | Annealing temperature | Restriction enzyme |
|---|---|---|---|---|
| *ephrinb1* | 5′-GTTTGTGTCTGGGAAGGGCTTAG-3′ | 5′-TATGGTGCTGCAGGACTCGGCCTG-3′ | 65 °C | XcmI (mutant: 2 fragments) |
| *ephb3b* | 5′- GGCCTGTTGTTTCCTCAGACT-3′ | 5′- GCAAACATGACCTTGTTGAGA-3′ | 65 °C | XcmI (mutant: 2 fragments) |
| *ephrinb2a* | 5′- TTTTGATCTAGAGAGAAATGCGAGT-3′ | 5′- TAGAGGCGTGTCTGCTTTTGACACCTG-3′ | 53 °C | PshAI (mutant: 2 fragments) |
| *ephb4a* | 5′- AGGCGGAGAGAAGAAAGTCAA-3′ | 5′- TGTTACCTGCATTGCCAAAGG-3′ | 56 °C | AlwnI (mutant: 2 fragments) |

**Genotyping of genetic mutants**. Adult mutant carriers are identified by PCR and restriction digest-based genotyping of tailfin tissue. To genotype embryos for experiments, we collected 1–2 mm of tail tissue after fixation to determine the genetic status by PCR and restriction digest. Conditions for PCR and restriction digests are shown in Table 2.

**Immunohistochemistry**. Embryos are fixed with 4% paraformaldehyde (PFA) overnight at 4 °C, permeabilized with PBST (PBS with 0.1% Triton x-100) and deyolked, followed by blocking with 1% Triton-100 and 10% donkey/goat serum in PBS for 1 h and incubated with primary antibodies, α-2F11 (mouse monoclonal, 1:1000, gift from Julian Lewis), α-ZO1 (mouse monoclonal antibody, 1:200, Invitrogen, 33–9100), aPKCζ (rabbit polyclonal antibody,1:1000, Santa Cruz, sc-216), α-pan-Cadherin (mouse monoclonal antibody, 1:1000, Sigma, C1821, rabbit α-EphrinB1[30] and guinea pig α-EphB3b[30] overnight at 4 °C. After repeated washing with PBST, embryos are incubated in secondary antibodies and fluorescent stains, donkey-anti-rabbit Cy3 (1:200, Jackson Immunoresearch, 112-166-143), donkey-anti-mouse 647 (1:400, Jackson Immunoresearch, 715-605-151), donkey-anti-goat 488 (1:200, Jackson Immunoresearch, 705-545-147) and Alexa Fluor 488 Phalloidin (1:500, Molecular Probes, A12379), Alexa Fluor 647 Phalloidin (1:500, Molecular Probes, A22287), DAPI (1:1000, Sigma, D9542) overnight at 4 °C. For nuclear protein detection with α-Prox1 (mouse monoclonal antibody, 1:50, Abcam, ab33219), rabbit polyclonal antibody, 1:500, Angiobio, 11-002P) and α-Hnf4a (goat polyclonal antibody, 1:100, Santa Cruz, sc-6556), embryos are incubated with DNaseI enzyme at 37 °C for 45 min prior to blocking[65] and washed with PBS, followed by normal staining protocol. To detect phospho-MLC we followed Sidhaye et al.[66], we permeabilized embryos with PBST (PBS with 0.8% Triton X-100) and trypsinized on ice for 15 min, followed by blocking with 10% goat serum and incubation in α-pMLC (rabbit Ser19; 1:100; Cell Signaling, #3671S) 1% goat serum in PBST for 60 h at 4 °C. After washing, the secondary antibody mix with 1% goat serum in PBST is added for 60 h.

Polyclonal antibodies against EphrinB2a and EphB4a were generated in rabbits and guinea pigs, respectively[30]. We subcloned the ectodomain region with the native signal peptide of zebrafish EphrinB2a and EphB4a into a mammalian expression vector containing a C-terminal rat Cd4d3 + 4 tag and a hexa-his tag for expression of a soluble recombinant protein by transient transfections into HEK293E cells (Bushell et al., 2008). Ni$^{2+}$-affinity chromatography (GE Healthcare) was used for protein purification. The animal facility contracted to produce antibodies is based in the UK and governed by UK legislation including the Animal (Scientific Procedures) act 1986. This facility was regularly inspected by the UK Home Office and all works subject to an ethical review.

Embryos were mounted either in 4% agarose and sectioned 130 μm using a Leica Vibratome or as whole-mount in Benzyl Alcohol, Benzyl Benzoate (BABB) clearing mixture prior to imaging. The embryos were imaged with a Leica SP8 or a Zeiss 780 confocal microscope. Images were processed with Imaris (Bitplane) image analysis software and Adobe Photoshop CS.

**Heat-shock experiment**. Embryos collected from *Tg(UAS: ephrinb1$^{EC}$)$^{nim25}$* and *Tg(hsp70l:Gal4)$^{fci1}$* intercrosses were subjected to heat shocks at 39 °C: 45 min either at 50 hpf or 54 hpf. The embryos were then returned to 28 °C incubator until fixation at 60 hpf.

**Mosaic cell labeling by DNA injection**. Membrane-targeted lyn-tdTomato under the control of the ubiquitin (ubi) promoter was co-injected at 20–30 pg DNA/embryo with 30–80 pg/embryo *transposase* mRNA into 1-cell-stage *Tg(Xla.Eef1a1: GFP)$^{s854}$* embryos.

**Blebbistatin treatment**. Myosin II ATPase activity was inhibited by blebbistatin[67]. Considering the impact of actomyosin dependent contractility in various cellular processes, we used mild doses of the drug (65 μM of final concentration in embryo water). Embryos were incubated from 52 hpf until 60 hpf at 28 °C under slow shaking conditions to prevent precipitation of the solution. This treatment caused no apparent change to the overall embryo body phenotype.

**Quantification of phenotype severity**. The variability of the mutant lumen formation defects was quantified by scoring ductal malformations in a domain-specific fashion. Each domain of the HPD system was assessed individually on a scale from 0 to 2, with 0 = no defect; 1 = mild defect; 2 = strong defect (Supplementary Fig. 3a–f, Table 1). 3D-confocal stacks of aPKC immunostainings of control and mutant HPDs were renamed for blind analysis and independently scored by two authors. The scores were independently analyzed with the Fisher Exact Test for frequency data. The robustness of the scoring was assessed by comparing and averaging both sets of scores. Given they are frequency data, they were first converted into ordinal data and then tested with the Wilcoxon test for non-parametric data (Supplementary Fig. 3g–j).

**Cell shape analysis**. To assess CBD cell shapes, the length of apical and basal cell domains and cell height were measured in embryos, in which aPKC labeled the apical membrane, panCadherin outlined cells and nuclear Dapi served to identify individual cells. 3D-stacks of confocal images were processed in 2D focal sections with Imaris (Bitplane) analysis software and cellular parameters were determined with the Measurement plug-in. Cells with unclear boundaries or complex membrane topologies possible due to contributions from multiple cells were excluded from the analysis. Statistical significance was assessed by unpaired Student's *t*-test.

**EdU incorporation assay and proliferation analysis**. Embryos were incubated for 60 min at 28.5 °C in 400 μM 5-ethynyl-2'-deoxyuridine (EdU) with 15% DMSO in egg water, and control larvae were treated with 15% DMSO. Larvae were fixed after incubation for 90 min at room temperature and processed using the Click-iT EdU Imaging Kit (Invitrogen). EdU incorporation was quantified with Imaris analysis software. Anxa4 antibody staining was used to mask the HPD ducts via the surface tool. DAPI- and EdU-positive cells were counted in the masked channels by the spot function and the ratio of EdU- to DAPI-positive cells was calculated subsequently.

**Quantification of pMLC levels**. Wild-type and *ephrinb1* mutant embryos stained for pMLC, ZO-1 and actin were used for pMLC quantification. Using FIJI[68], for each sample three linear ROIs where drawn perpendicular to the duct luminal axis and on a single plane, in which the lumen was clearly visible; each ROI was 10 pixels/1.9 μm wide and spanning the whole tube across opposite basal sides (52.7 μm). Average intensities in each position of the line were extracted using the Plot Profile plugin in FIJI. A rectangular ROI of 40 μm$^2$ was used to extract the average background intensity in each channel and used to normalize the fluorescence intensity values within the linear ROIs. Two regions of 2 μm each were selected across the apical peaks in the intensity profile of the ZO-1 channel and the background-corrected pMLC values where integrated in the chosen areas for each linear ROI (see Supplementary Fig. 5).

**Microarray analyses of human tissue samples**. Preprocessing of microarray data from LEC2012[69], LEC2018 and normal bile duct (NBD) include background correction, quantile normalization and log$_2$-transformation of intensity values. Differentially expressed genes were calculated by t-test analysis using R v3.3. All samples were obtained with approval by the institutional review board of the National Institutes of Health and collaborating institutions on the condition that patients were anonymized[69] and subsequently approved by the Danish ethics committee.

**Reporting summary**. Further information on research design is available in the Nature Research Reporting Summary linked to this article.

## Data availability

The authors declare that all data supporting the findings of this study are available within the article and its supplementary information files or from the corresponding author upon reasonable request. The image data generated and analyzed in this study are available from the corresponding author upon reasonable request. A source data file for

the quantifications shown in Figs. 1g, 2h, o, p, 3e–g, 4e, h, 7f, l, m, p and Supplementary Figs. 3g–j and 6d, e are provided as a Source Data file.

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

## Acknowledgements

We thank A. Vanoosthuyse for expert technical support, J. Sedzinski, R. Thiagarajan, A. Grapin-Botton, I. Salecker, O. Anderson, and the Ober group for continued discussions or comments on the manuscript, J. Bulkescher and J. Dreyer (DanStem Imaging core facility) for assistance with image acquisition and analysis, and the department of experimental medicine (AEM) for expert fish care. This work was funded by the Novo Nordisk Foundation (NNF17CC0027852) and Danish National Research Foundation (DNRF116). J.C. and D.G. W. were supported by the Francis Crick Institute, which receives its core funding from Cancer Research UK (FC001217), the UK Medical Research Council (FC001217), and the Wellcome Trust (FC001217). S.C. was supported by an SNSF Early Postdoc Mobility fellowship (P2ZHP3_164840) and a Long Term EMBO Postdoc fellowship (ALTF 511-2016), and L.S. and J.B.A. by the Independent Research Fund Denmark (DFF; Sapere Aude2 4183-00118B).

## Author contributions

M.I.T., S.C., and E.A.O. conceived the project. M.I.T., S.C., J.C., R.S.L.H., and R.A. performed the experiments and analyzed the data. W.H. performed statistical analysis. L.S. and J.B.A. contributed the human tissue expression data. M.I.T and E.A.O. wrote the manuscript with support from all authors.

## Competing interests

The authors declare no competing interests.
