## [Peer Review File · Nature Communications]

Reviewers' Comments:

Reviewer #1:

Remarks to the Author:

Thestrup M.I et al. beautifully show how the lumen in the HPD system forms through ductal morphogenesis and identify EphB/EphrinB signaling as a critical regulator of the morphogenesis. These findings are highly novel, making this manuscript significant. I have following comments that need to be addressed to further improve the presented work.

- aPKC and F-actin staining reveals the polarity of ductal cells but not a lumen. Given the emphasis of lumen formation in the manuscript, it is better to show a lumen and a microlumina with hollow space by 3-D rendering, EM, or section staining.
- Although criteria for HPD phenotypic scoring are described (Fig. S2), such classification is rather ambiguous. This ambiguity might lead to a wrong interpretation of double mutant phenotypes shown in Fig. 6F. For example, EPD phenotype in ephrinb1;ephrinb2a double mutants appears to be less severe than that in ephrinb1 or ephrinb2a single mutants. Is it true? If so, please speculate why that happens. If not, statistical significance has to be marked in such phenotype strength graphs. In addition, is it sufficient to claim a lumen just based on aPKC staining (Fig. S2)?
- It is also written that ephb4a mutants showed no detectable change in the HPD at 60 hpf and 5 dpf (line 385). But Fig. 6K image shows much shorter CD and CBD in ephb4a mutants than in controls.
- Given the promiscuity of ephrin/Eph ligand-receptor binding, it may be worth analyzing ephrinb1;ephb3b double mutants and compare their HPD phenotypes with ephrinb1 and ephb3b single mutants. ephrinb1 can activate other ephb receptors; ephb3b can activate other ephrinb ligands. Thus, much severe defects might be exhibited in the double mutants.
- Given the speculation that EphB/ephrinB signaling controls HPD morphogenesis through actomyosin, it is better to provide any genetic evidence supporting the speculation. For example, blebbistatin treatment with a suboptimal dose may induce HPD defects in ephb3b or ephrinb2a het but not in wild-type embryos.
- Human data presented in Fig. 5C are not relevant to the zebrafish HPD study, because the human samples are intrahepatic ducts, not extrahepatic ducts. Human EHD, GB, CBD, CD, or EPD tissues should be used.
- There is a typo in line 378: Ephb2a should be ephrinb2a.

Reviewer #2:

Remarks to the Author:

Report on "A morphogenetic EphB/EphrinB code controls hepatopancreatic duct formation » by Thestrup et al.

This article presents a detailed description of the morphogenesis of the hepatopancreatic ductal (HPD) system in zebrafish and reveals the role of several ephrinB ligands and ephB receptors in different steps of this process as well as in different compartments.

The authors identify key steps of HPD morphogenesis by high resolution confocal microscopy with a set of epithelial and apical markers. The process starts with the appearance of junctional aggregates in the unpolarised epithelium then de novo lumen formation through fusion of disconnected microlumina and remodelling of the nascent ducts. By chemical approach, they show that epithelial cell rearrangement during HPD remodelling requires non-muscle myosin II activity.

Through mutant loss-of-function and transgenic overexpression approaches, they also uncover important functions for ephrinb1, ephrinb2a and ephB3b (and to a minor extent ephB4) in HPD remodelling and lumen resolution in different parts of the HPD. These ligands and receptors are expressed in partially overlapping domains of the HPD which translates into regionalized Eph/Ephrin activity within the HPD, supporting the existence of an "Ephrin/Eph code".

Overall, the study is original as it uncovers the process of HPD morphogenesis which was so far poorly understood. This is of significant interest as the HPD may be the site of human diseases. The study also convinces that Ephrin/Eph are key novel actors in HPD formation and it brings some interesting elements to the mechanism by which these proteins control HPD morphogenesis. However, some presented data don't sufficiently support several conclusions and more mechanism is required. Below I identify several weaknesses and aspects that should be improved and/or investigated further to consolidate the impact of the study.

One major weakness is the quality of some images which is not convincing enough, particularly in the mutants. Indeed, some pictures of the HPD system are really difficult to interpret, most notably for the mutants in Fig 2H (72 hpf) but also for wild type samples in Fig 1 E-E'. For instance, it is hard to evaluate if real ductal loops are shown while loop-like structures may result from projection into one view of different independent elements stacked together (see notably Fig 1B'). In addition, these views are often mixed with the intestine below which does not simplify the analysis. This leads the reader to rely on the cartoons to understand the confocal images, which is normally not the purpose of such schemes. Higher magnification, 3D rendering and/or short videos could help improve the visualization.

Consequently, the presented images don't always convince about more mechanistic aspects of epithelial remodelling occurring in the mutants. For example, it is concluded that ephrinB/ephb control HPD morphogenesis through cell rearrangement by intercalation based on the observation of "more frequent" teardrop-shaped cells in mutant HPD. While teardrop-shaped cells are easy to distinguish in Supplemental Fig 3B in the case of WT embryos, the cells that are indicated in the mutants in Fig 3C" and D" are barely recognizable. These illustrations fail to convince that cell rearrangement through intercalation requires Ephrin activity. Unambiguous identification of these cells, as well as formal quantification in WT versus mutant would strengthen this conclusion.

Another caveat is that the processes described in the study are very dynamic and depend on active cell rearrangement. Although I recognize the efforts made to present the different steps of HPD morphogenesis with different markers at different time points, all views are static and our understanding of the phenomenon would be greatly improved by in vivo imaging (if feasible with such samples).

Finally, the mechanisms by which ephrinb/ephb control HDP morphogenesis should be explored more carefully and more deeply. For example, the HPD undergoes a phase of extension between 52 and 60 hpf. This extension is compromised in the mutants and it is concluded that impaired cell rearrangement in ephrinb1 and ephb3b mutants results in failure of CBD extension. This is assumed from the presence of more abundant teardrop-shaped cells in the mutants and by some similarities between the mutant phenotype and defects obtained after treatment with the myosin II inhibitor blebbistatin. First, I am not convinced based on the presented data that mutants really harbour more teardrop-shaped cells. Indeed, lines 244-245 refer to Supplemental Figure 3 simply showing wild type HPD views while images and quantifications of both WT and mutants would be expected from the text. This is misleading. There should be a formal quantification of the "increased number" of teardrop-shaped cells. Second, as discussed above, some pictures of the mutants are not enough convincing. Third, it should be discussed why the possibility of other defects such as cell number alteration in the mutants is not considered to explain the phenotype. Finally, a clear molecular link between EphrinB signalling and the ductal phenotype remains to be established. Additional analyses such as pMLC in the mutants should be shown.

Membrane protrusions pointing toward forming lumina are presented in Supplementary Figure 3 in wild type embryos. To further elucidate how EphrinB/EphB function in tube formation, similar analysis could also been done in mutant HPD or in dominant negative transgenic embryos with quantification of the orientation of the protrusions.

Finally, as some ephrin ligands and receptors are not only expressed in the endoderm but also in

the mesoderm surrounding the HPD, elucidation of their function in both compartments respectively would greatly strengthen the concept of EphrinB/EphB code.

Minor points:

Line 92: this is not in vivo analysis.

Fig 3B'-D" show lumen defects in the common bile duct of the mutants. In fact, it seems that mutant CBD display multiple, maybe disconnected, lumina. What precisely is quantified in Fig 3E? The addition of several individual lumen length? This should be clarified.

In the "EphB/EphrinB signaling is specifically required for lumen remodeling in HPD morphogenesis" Result section, a quite large part of the paragraph is dedicated to show and discuss the ectopic structures observed upon overexpression of a dominant negative form of ephrinB. Although interesting, this could be shortened and more focused on the 2 main messages : i) ephrinB signalling required specifically in HPD morphogenesis and ii) possible involvement of additional EphrinB and EphB.

Line 238+249 : the text refers twice to Supplemental Figure 3 A-A' for both teardrop-shaped cells and protrusions.

Line 353 : defective endocytosis in EphB3b mutant is mentioned to explain extension of EphrinB1 protein domain. This explanation does not consider possible ectopic expression of ephrinb1 mRNA, which would also be plausible.

Line 354 : please cite all candidate ephrin and eph tested for expression analysis.

Line 378 : "given Ephrinb1 and EphB2a coexpression..." correct EphB2a to EphrinB2a.

Generation of genetic mutants : a schematic representation of the targeted genomic sequence and the resulting mutant protein could be shown in supplementary data.

Reviewer #3:

Remarks to the Author:

In their manuscript entitled "A morphogenetic EphB/EphrinB code controls hepatopancreatic duct formation" Thestrup et al. use high resolution imaging in zebrafish to define a critical period during zebrafish development when the hepatopancreatic ductal system forms through de novo lumen formation in a process that requires myosin contractility. Using a candidate approach the authors identify a role for different ephrin ligand/receptor pairs in duct formation and gallbladder size.

Overall, this is a well-executed study described in exceptional clarity with convincing microscopy and accompanying quantitative analyses. The use of (existing and newly generated mutants), inducible transgenic constructs and chemicals gives compelling support for the proposed mechanisms. The discussion is balanced and does highlight further areas of investigation. There are no significant criticisms or experimental shortcomings.

Reviewer #4:

Remarks to the Author:

In this manuscript, Thestrup et al, demonstrate hepatopancreatic duct (HPD) formation from a solid cluster of primordial endodermal cells via stepwise hollowing out process. Interesting phenotypes were characterized in detail for EphB3b and EphrinB1 mutants, and additional report on the phenotypes of EphrinB2a and EphB4a, as well as the combinations thereof. The spatiotemporal expression and distribution of the two EphB receptors and two EphrinB ligands further support the phenotypic observations. These studies clearly established important roles of the EphB/EphrinB directional signaling in the genesis of HPD, and are quite significant. Several comments are listed below.

1) It is apparent that the deleterious effects resulting from EphrinB1 and EphB3 mutations were already done by 52 hpf. Earlier time points, e.g., 46 or 48 hpf could be examined in greater details, including the expression pattern.

- 2) Since EphB/EphrinB interactions also regulates cell proliferation, it EphrinB1 and EphB3 deletion may cause aberrant proliferations in the wrong place or at wrong time.
- 3) Please explain the "phenotype" in WT embryos in the quantitative analysis of the WT, such as 10% severe phenotype in CBD (Fig. 2G).
- 4) Figure 3F,G, the effects of EphB3b and EphrinB1 mutation on the actomyosin were not described. Are EphB3 and Ephrin-B1 linked to actomyosin regulation?

Other Points

- 1) In the introduction, a sentence between lines 50-51 can be moved to line 45 after "HPD", to indicate that the structure detailed in the introduction refers to fish. There are differences between mammalian and zebra fish HPD.
- 2) Instead of 2F11 in Fig 1 legend and line 212, use Annexin A4. It is unnecessarily confusing. Define XlaEef1a1:GFPs854 for the readers.

Point by point response to editorial and reviewers' comments

We thank the reviewers for their constructive comments, which have helped us to improve our study. To address the raised points, we have made changes to the text and carried out more experiments, including (i) a quantification of cell shapes (ii) the expression and localization of pMLC, (iii) proliferation analysis and (iv) analysis of HPD formation at earlier stages in *ephrinb1*, *ephb3b* embryos and controls to elucidate the mechanism by which they contribute to HPD remodeling. Moreover, we have improved the visualization of the phenotypes and developed a more robust scoring system for easier access of our data.

Reviewer #1:

1 - aPKC and F-actin staining reveals the polarity of ductal cells but not a lumen. Given the emphasis of lumen formation in the manuscript, it is better to show a lumen and a microlumina with hollow space by 3-D rendering, EM, or section staining.

We agree with the reviewer that neither aPKC nor F-actin staining indicate a patent open lumen. Both F-actin and aPKC are however well known to localize to the apical domain of cells (Datta et al. 2011) lining the lumen of epithelial tubes in most ductal systems. Moreover, aPKC is required in many cases for lumen formation across species (Baer and Affolter, 2009). For these reasons, aPKC is a good readout of progressing lumen and tube formation.

To address this concern, we generated optical cross-sections of 5 dpf ducts labeled with Anxa4, Phalloidin and DAPI. This revealed open lumina devoid of nuclei, indicative of patent ducts at this stage (added to Figure 1). This is in line with functional assays using secreted proteins to test for a functional ductal system (Farber et al. 2001). Furthermore, we clarified in the text (lines: 146-147) that aPKC staining at stages before 5 dpf does not necessarily indicate open lumens. In addition we altered the text throughout the manuscript to indicate that aPKC at examined stages labels nascent lumina.

2 - Although criteria for HPD phenotypic scoring are described (Fig. S2), such classification is rather ambiguous. This ambiguity might lead to a wrong interpretation of double mutant phenotypes shown in Fig. 6F. For example, EPD phenotype in *ephrinb1;ephrinb2a* double mutants appears to be less severe than that in *ephrinb1* or *ephrinb2a* single mutants. Is it true? If so, please speculate why that happens. If not, statistical significance has to be marked in such phenotype strength graphs. In addition, is it sufficient to claim a lumen just based on aPKC staining (Fig. S2)?

We agree with the reviewer that scoring of the HPD phenotypes in the mutants requires objective criteria. As adequate comparisons of HPD defects between genotypes are essential, we introduced a more robust and simplified scoring system, using 0 (no), 1 (weak) and 2 (strong) defects instead of a 0-3 scale (Supplementary Figure 3a-f), with criteria described in Table1. Results from this simplified scoring system are shown in Figures 2h and 7f. Moreover, we increased sample sizes to give the analysis more statistical power. All samples were assessed blinded and independently scored by two authors, producing two highly similar scores. Significance for the individual data sets was determined by a Fisher Exact test for frequency data. To allow averaging of both sets of frequency data, their scores were converted into ordinal values and then assessed by the non-parametric Wilcoxon test. Its outcome generally confirmed the significance values determined for the single data sets (Supplementary Figure 3g-j).

Concerning the severity of EPD defects in *ephrinb1;ephrinb2a* double mutants compared to either single mutants, rigorous scoring of an increased sample number and statistical analysis showed that there is a mild phenotype in a subset of *ephrinb1* single and *ephrinb1;ephrinb2a* double mutants and thus no rescue.

Finally, aPKC staining is per se not sufficient to describe open lumens, which is discussed in the response to point 1.

3 - It is also written that *ephb4a* mutants showed no detectable change in the HPD at 60 hpf and 5 dpf (line 385). But Fig. 6K image shows much shorter CD and CBD in *ephb4a* mutants than in controls.

Thank you for the keen observation. The more robust scoring confirmed no significant changes in the formation of the individual HPD domains at 60 hpf. We measured the CBD in all mutants at 5 dpf, revealing no significant change in *ephb4a* compared to

controls. However, it is significantly shorter in *ephrinb2a* and *ephb3b*.

4 - Given the promiscuity of ephrin/Eph ligand-receptor binding, it may be worth analyzing *ephrinb1;ephb3b* double mutants and compare their HPD phenotypes with *ephrinb1* and *ephb3b* single mutants. *ephrinb1* can activate other *ephb* receptors; *ephb3b* can activate other *ephrinb* ligands. Thus, much severe defects might be exhibited in the double mutants.

To address this valid point, we analyzed HPD morphology at 60 hpf in *ephrinb1;ephb3b* mutants compared to single mutants. This resulted in three main conclusions: (i) phenotypes in double *ephrinb1;ephb3b* occur with the same frequency and severity in most parts of the HPD compared to *ephb3b* mutants, revealing no additive or synergistic effects. (ii) malformations in *ephrinb1;ephb3b* double mutants are significantly more frequent and/or more severe than those in *ephrinb1* mutants, suggesting that likely an additional EphrinB ligand interacts with EphB3b. Consistent with this observation, the phenotype frequency and severity in *ephrinb1;ephb3b* are highly similar to defects detected in *ephrinb1;ephrinb2a* double mutants (Figure 2, 7, Supplementary Figure 3). This applies in particular to the EHD, GB and CBD, supporting the notion that both EphrinB1 and EphrinB2a act in a partially redundant manner in these structures. This further suggests that a specific level of ligand is necessary to ensure robust duct remodeling. (iii) *ephrinb1;ephb3b* mutants show a lower statistical significance of strong phenotypes in the gall bladder than *ephb3b* single mutants, likely due to smaller sample number, or alternatively suggesting a potentially antagonistic interaction.

5 - Given the speculation that EphB/ephrinB signaling controls HPD morphogenesis through actomyosin, it is better to provide any genetic evidence supporting the speculation. For example, blebbistatin treatment with a suboptimal dose may induce HPD defects in *ephb3b* or *ephrinb2a* het but not in wild-type embryos.

This is an important point, which was also raised by reviewers 2 and 4. We sought to address the relationship of EphB/EphrinB and actomyosin activity in two ways. For these experiments we focused on *ephrinb1* mutants as they exhibit the most prominent changes in cell shape, First, employing a genetic enhancer/suppressor strategy, heterozygous *ephrinb1* embryos were treated with reduced doses of blebbistatin (0,5x and 0,25x) during the remodeling phase (52-60 hpf), resulting in low frequency disruption

of duct formation in both controls and *ephrinb1* +/- embryos at similar frequency. As a single copy of *ephrinb1* may not necessarily generate a sensitized background, this set of experiments was therefore inconclusive.

In a second set of experiments we stained *ephrinb1* mutants for pMLC at 60 hpf. Quantification of pMLC levels in the CBD by staining intensity revealed a significant increase of pMLC at the apical domain of up to 6-fold in *ephrinb1* mutants. Specifically, we have examined pMLC localization in the CBD in longitudinal sections and measured pixel intensities of the apical domain in multiple CBD positions per embryo. This suggests that EphrinB1 could directly control pMLC localization to the apical side, or indirectly for instance due to the altered cell shapes.

In the course of these experiments we have used a different and consistent staining protocol, and therefore replaced the previous panels; both protocol and antibody source are described in the Materials and Methods section.

6 - Human data presented in Fig. 5C are not relevant to the zebrafish HPD study, because the human samples are intrahepatic ducts, not extrahepatic ducts. Human EHD, GB, CBD, CD, or EPD tissues should be used.

Indeed the best way forward would have been to analyze healthy human HPD tissue. However, given the importance of the ductal system to body physiology and its delicate structure, it is not possible to obtain human biopsies. For this reason we had chosen to check EPHRIN and EPH expression in large intrahepatic ducts. These are closest to the hilar ducts and macroscopically dissected by a skilled surgeon. Strikingly, we find that tissue from human cholangiocarcinoma of the common bile duct (encompassing the duct between liver and the HPA, which corresponds to the EHD and CBD in zebrafish) shows very similar ligand and receptor expression. This suggests that EPHRINB ligand and EPHB receptor could be similarly expressed in human extrahepatic ducts. However, as this represents indirect evidence, we have updated the text, lines 307-311, and moved the data to Supplementary Figure 8.

7 - There is a typo in line 378: Ephb2a should be ephrinb2a.

Thank you for pointing this out – it has been changed.

Reviewer #2:

1 - One major weakness is the quality of some images which is not convincing enough, particularly in the mutants. Indeed, some pictures of the HPD system are really difficult to interpret, most notably for the mutants in Fig 2H (72 hpf) but also for wild type samples in Fig 1 E-E'. For instance, it is hard to evaluate if real ductal loops are shown while loop-like structures may result from projection into one view of different independent elements stacked together (see notably Fig 1B'). In addition, these views are often mixed with the intestine below which does not simplify the analysis. This leads the reader to rely on the cartoons to understand the confocal images, which is normally not the purpose of such schemes. Higher magnification, 3D rendering and/or short videos could help improve the visualization.

We appreciate the comments of the reviewer. To improve the visualization of ductal morphologies, most of the panels now show projections of only the focal planes including the HPD and not the entire digestive system (such as the intestine). In more challenging time series the HPD is highlighted by color for easier distinction. In addition, short movies showing the HPDs of controls and mutants at 60 hpf and 5 dpf by aPKC or Anxa4 staining, respectively, are added as Supplementary Movies 1-13.

2 - Consequently, the presented images don't always convince about more mechanistic aspects of epithelial remodelling occurring in the mutants. For example, it is concluded that *ephrinB/ephB* control HPD morphogenesis through cell rearrangement by intercalation based on the observation of "more frequent" teardrop-shaped cells in mutant HPD. While teardrop-shaped cells are easy to distinguish in Supplemental Fig 3B in the case of WT embryos, the cells that are indicated in the mutants in Fig 3C" and D" are barely recognizable. These illustrations fail to convince that cell rearrangement through intercalation requires Ephrin activity. Unambiguous identification of these cells, as well as formal quantification in WT versus mutant would strengthen this conclusion.

To address this important point, we analyzed cell shapes in the CBD of control and *ephrinb1* and *ephb3b* mutants by staining the membranes with α -panCadherin to outline duct cells and aPKC to indicate the apical side. To quantify cell shape changes, the apical and basal domain were measured. We subsequently calculated the basal/apical ratio for each cell, revealing a clear increase of tear-drop shaped cells facing the forming lumen in both mutants (Figure 3f). This increase occurs at the expense of tear-drop

shaped cells facing away from the lumen and columnar cells. In addition, ductal cells are higher in particular in *ephrinb1* mutants compared to controls (Figure 3g), supporting the idea that duct formation and extension occurs by active rearrangement and cell intercalation. Furthermore, they indicate that these cell shape changes require EphB/EphrinB signaling. To clarify this point, we have highlighted representative examples of cell shapes in Fig. 3.

3- Another caveat is that the processes described in the study are very dynamic and depend on active cell rearrangement. Although I recognize the efforts made to present the different steps of HPD morphogenesis with different markers at different time points, all views are static and our understanding of the phenomenon would be greatly improved by in vivo imaging (if feasible with such samples).

We agree with the reviewer that *live* imaging is the most appropriate way to investigate dynamic processes and have put significant effort into establishing this approach for the HPD. Despite testing different strategies, this was so far unsuccessful. The main limiting factors are (i) the lack of appropriate transgenic lines driving gene expression specifically in the HPD and (ii) its position within the embryo. The only lines with expression in the HPD are the pan-endodermal *tg(XlaEef1a1:GFP)^{s854}* and the *tg(keratin18:EGFP)^{p314}*. Both lines are unsuitable for studying epithelial cell behaviors, since GFP is cytoplasmic and not membrane-tethered. Mosaic labeling with membrane-tethered fluorescent proteins by DNA injection resulted in too low expression to study *live* cell behaviors. Moreover, in the course of this study we also found that the *hsp70l* promoter, commonly used for conditional gene expression, does not activate expression in the HPD. We tested more than five transgenic lines for tagged-polarity or membrane proteins controlled by the *hsp70l* promoter, including the *hsp70l:gal4* line. Blocking of EphB/EphrinB signaling by conditional expression of the secreted EphrinB1^{EC} is possible, because the *hsp70l:gal4* is active in the surrounding mesenchyme, the liver and pancreas.

Finally, imaging with a regular and multi-photon confocal microscope only produced weak and low-quality images despite the transparency of zebrafish because the forming HPD is located deep in the embryo. It is in the least suitable position for imaging in terms of actual distance and location next to the yolk causing light scattering.

4 - Finally, the mechanisms by which *ephrinb/ephb* control HPD morphogenesis should be explored more carefully and more deeply. For example, the HPD undergoes a phase of extension between 52 and 60 hpf. This extension is compromised in the mutants and it is concluded that impaired cell rearrangement in *ephrinb1* and *ephb3b* mutants results in failure of CBD extension. This is assumed from the presence of more abundant teardrop-shaped cells in the mutants and by some similitudes between the mutant phenotype and defects obtained after treatment with the myosin II inhibitor blebbistatin. First, I am not convinced based on the presented data that mutants really harbour more teardrop-shaped cells. Indeed, lines 244-245 refer to Supplemental Figure 3 simply showing wild type HPD views while images and quantifications of both WT and mutants would be expected from the text. This is misleading. There should be a formal quantification of the “increased number” of teardrop-shaped cells.

Following the reviewer’s suggestion, a formal quantification was performed showing a clear change in cell shape distribution in both *ephrinb1* and *ephb3b* mutants. Please, see also our response to point 2.

5 - Second, as discussed above, some pictures of the mutants are not enough convincing.

Our strategy to improve HPD visualization in controls and mutants is outlined in our response to point 1.

6 - Third, it should be discussed why the possibility of other defects such as cell number alteration in the mutants is not considered to explain the phenotype.

Changes in cell number and proliferation rate could indeed contribute to the observed phenotypes, as EphrinB/EphB signaling regulates proliferation in some tissues (Kania and Klein, 2016) (see also point 2 of reviewer 4). To determine proliferation rates during HPD formation, we performed EdU incorporation experiments during early duct remodeling in *ephrinb1* and *ephb3b* mutants. They showed an albeit not statistically significant mild increase of EdU⁺ cells in *ephrinb1* mutants compared to controls. Moreover, the overall cell number of Anxa4/2F11⁺ HPD cells is similar between controls and mutants corroborating that changes in cell number have no or little contribution; these data are added as Supplementary Fig. 6.

7 - Finally, a clear molecular link between EphrinB signalling and the ductal phenotype remains to be established. Additional analyses such as pMLC in the mutants should be shown.

To investigate a possible link between EphrinB/EphB signaling and myosin, which was also stressed by reviewer 1 and 4, we have examined the interaction in two ways focusing on *ephrinb1* mutants as they exhibit the most prominent changes in cell shape. First, employing a genetic enhancer/suppressor strategy, heterozygous *ephrinb1* embryos were treated with reduced doses of blebbistatin (0,5x and 0,25x) during the remodeling phase (52-60 hpf), resulting in low frequency disruption of duct formation in both controls and *ephrinb1 +/-* embryos at similar frequency. As a single copy of *ephrinb1* may not necessarily generate a sensitized background, this set of experiments was therefore inconclusive.

In a second set of experiments we stained *ephrinb1* mutants for pMLC at 60 hpf. Quantification of pMLC levels in the CBD by staining intensity revealed a significant increase of pMLC at the apical domain of up to 6-fold in *ephrinb1* mutants. Specifically, we have examined pMLC localization in the CBD in longitudinal sections and measured pixel intensities of the apical domain in multiple CBD positions per embryo. This suggests that EphrinB1 could directly control pMLC localization to the apical side, or indirectly for instance due to the altered cell shapes.

In the course of these experiments we have used a different and consistent staining protocol, and therefore replaced the previous panels; both protocol and antibody source are described in the Materials and Methods section.

8 - Membrane protrusions pointing toward forming lumina are presented in Supplementary Figure 3 in wild type embryos. To further elucidate how EphrinB/EphB function in tube formation, similar analysis could also been done in mutant HPD or in dominant negative transgenic embryos with quantification of the orientation of the protrusions.

We agree with the reviewer that assessing whether EphB/EphrinB signaling controls protrusion formation is an interesting point. We performed DNA injections for mosaic labeling of HPD cells with membrane-tethered mOrange in *ephrinb1* and *ephb3b* mutants to assess protrusion formation at 50 hpf. We were successful in labeling groups of ductal cells, but individual cells could not be distinguished within these because of

their clonal relationship (3 to 10 labeled clusters in ≥ 50 embryos per genotype) and labeled cell in adjacent mesodermal cells. Together this prevented any scoring of protrusions per individual HPD cell. Moreover, endoderm clones are generally rare (about 15-20%) and even less frequent in the small HPD. Similar to other experiments, an HPD-specific promoter for efficient activation of gene expression in the HPD would have been instrumental.

9 -Finally, as some ephrin ligands and receptors are not only expressed in the endoderm but also in the mesoderm surrounding the HPD, elucidation of their function in both compartments respectively would greatly strengthen the concept of EphrinB/EphB code.

This is an excellent point and particularly interesting for dissecting the role of EphB3b, which is expressed in the EPD endoderm and the mesoderm around the EHD, GB and CBD. As for *live* imaging experiments, a currently lacking transgenic driver for conditional gene expression in the HPD would be instrumental for such investigations.

To overcome this and address the reviewer's point at least in part, we initiated mosaic analysis by tissue-specific transplantation experiments (adopted from Stafford et al. 2006). For this, we transplanted endoderm-targeted *ephb3b*^{-/-} cells into wild type to test the requirement for EphB3b in the HPD endoderm (see controls and examples in the figure below, included only in this response), and endoderm-targeted wild type cells into *ephb3b*^{-/-} hosts to assess the requirement of EphB3b in the HPD surrounding mesoderm. Cell transplantation was overall successful, as determined at 24 hpf, however, the fluorescently-labeled dextran was strongly diluted and difficult to detect by 60 hpf, the time of analysis. This likely happens as a result of cell proliferation in the endoderm, and possibly metabolic processing and excretion of the dextran, since we see a dramatic increase of fluorescent puncta (Figure b''', e'''). In rare cases of *ephb3b*^{-/-} cells transplanted into wildtype we could clearly detect labeled *ephb3b*^{-/-} cells associated with a HPD defect (see results included below, Figure d-e'''), suggesting EphB3b is required for the formation of a single lumen tube. However, overall these experiments did not produce any clear conclusions.

Figure: Transplantation strategy and examples to assess tissue-specific functions of EphB3b.

Minor points:

Line 92: this is not in vivo analysis.

This is clarified in the text.

Fig 3B¹-D” show lumen defects in the common bile duct of the mutants. In fact, it seems that mutant CBD display multiple, maybe disconnected, lumina. What precisely is quantified in Fig 3E? The addition of several individual lumen length? This should be clarified.

We agree, the quantification has not been clearly described. For this quantification, the longest continuous aPKC staining/immature lumen was determined to describe the defect and is now specified in the figure legend..

In the “EphB/EphrinB signaling is specifically required for lumen remodeling in HPD morphogenesis” Result section, a quite large part of the paragraph is dedicated to show and discuss the ectopic structures observed upon overexpression of a dominant negative form of ephrinB. Although interesting, this could be shortened and more focused on the 2 main messages : i) ephrinB signalling required specifically in HPD morphogenesis and ii) possible involvement of additional EphrinB and EphB.

Following this suggestion, we have changed this part into a more concise paragraph.

Line 238+249 : the text refers twice to Supplemental Figure 3 A-A’ for both teardrop-shaped cells and protrusions.

This has been updated during these revisions

Line 353 : defective endocytosis in EphB3b mutant is mentioned to explain extension of EphrinB1 protein domain. This explanation does not consider possible ectopic expression of ephrinb1 mRNA, which would also be plausible.

We agree that ectopic expression of *ephrinb1* mRNA resulting from relieved repression is another possibility to explain the expansion of EphrinB1 expression in *ephb3b* mutants and have added this point to the text; lines 297-298.

Line 354 : please cite all candidate ephrin and eph tested for expression analysis.

We have tested specifically EphrinB2a and EphB4a given that they frequently act

together with EphrinB1 and EphB3b in intricate morphogenetic events (Kania and Klein, 2016), such as vascular morphogenesis, bone formation, or boundary formation in neural and non-neural tissues. Moreover, the availability of genetic mutants for both genes was important to ensure consistent analysis of late developmental phenotypes. This has been clarified in the text.

Line 378 : “given Ephrinb1 and EphB2a coexpression...” correct EphB2a to EphrinB2a.
Thank you for pointing this out - this has been changed.

Generation of genetic mutants : a schematic representation of the targeted genomic sequence and the resulting mutant protein could be shown in supplementary data.

A schematic depicting the *ephb3b* locus is now included in Supplementary Figure 1, including the gRNA targeting site in exon 2 and the resulting 4bp deletion and premature stop codon in the ligand binding domain of the *ephb3b*^{nim27} allele.

Reviewer #3:

No response is required.

Reviewer #4:

1) *It is apparent that the deleterious effects resulting from EphrinB1 and EphB3 mutations were already done by 52 hpf. Earlier time points, e.g., 46 or 48 hpf could be examined in greater details, including the expression pattern.*

To address this point, we stained *ephrinb1* and *ephb3b* mutants as suggested at 48 hpf with aPKC. Initiation of cell polarization and tube formation occur similarly to controls. aPKC positive structures are partly disorganized with disrupted and some parallel organisation (Supplementary Figure 2A-C), suggesting that both ligands and receptors are required from the beginning in the HPD primordium. In line with this conclusion, we detected EphrinB1 expression in the EHB, high EphB3b expression in the EPD and low expression in the EHB at this stage (Figure 6). Conditional inactivation of EphB/EphrinB uncovered a clear requirement in HPD tube formation. However, we cannot exclude

some contribution of the initial Prox1⁺ progenitor-positioning defect to the phenotype observed in the HPD primordium at 48 hpf, This is discussed in the last part of the manuscript.

2) Since EphB/EphrinB interactions also regulates cell proliferation, it EphrinB1 and EphB3 deletion may cause aberrant proliferations in the wrong place or at wrong time.

Changes in cell number and proliferation rate could indeed contribute to the observed phenotypes, as EphB/EphrinB signaling is known to regulate proliferation in some tissues (see also point 6 reviewer 2). To determine proliferation rates during HPD formation, we performed EdU incorporation experiments during early duct remodeling in *ephrinb1* and *ephb3b* mutants. They showed a mild, however not statistically significant, increase of EdU⁺ cells in *ephrinb1* mutants compared to controls. Moreover, the overall cell number of 2F11⁺ HPD cells is similar between controls and both mutants corroborating that changes in cell number have no or little contribution; these data are added as Supplementary Figure 6.

3) Please explain the “phenotype” in WT embryos in the quantitative analysis of the WT, such as 10% severe phenotype in CBD (Fig. 2G).

Our data indicate that 10% of wild type larvae displayed defects in the CBD. These can be strong and include the incorrect formation of for instance a large loop or expanded/cyst-like aPKC-positive domain. We believe this reflects some inherent variability of this complex process. For this reason, we have revisited the scoring and to make it more robust added more samples, simplified the scoring to a 0-2 instead of a 0-3 scale, and had two authors independently score the samples blindly and perform statistical analysis (for details please see comments to point 2 reviewer 1, Material and Methods, Table 1 and Supplementary figure 3). Moreover, we thank the reviewer for picking up the inaccurate use of the word ‘phenotype’ in this context, we altered it throughout the text.

4) Figure 3F,G, the effects of EphB3b and EphrinB1 mutation on the actomyosin were not described. Are EphB3 and Ephrin-B1 linked to actomyosin regulation?

We agree with the importance of this point, which was also raised by reviewer 1 and 2.

To investigate a possible link between EphrinB/EphB signaling and myosin, which was also stressed by reviewer 1 and 4, we have examined the interaction in two ways focusing on *ephrinb1* mutants as they exhibit the most prominent changes in cell shape. First, employing a genetic enhancer/suppressor strategy, heterozygous *ephrinb1* embryos were treated with reduced doses of blebbistatin (0,5x and 0,25x) during the remodeling phase (52-60 hpf), resulting in low frequency disruption of duct formation in both controls and *ephrinb1 +/-* embryos at similar frequency. As a single copy of *ephrinb1* may not necessarily generate a sensitized background, this set of experiments was therefore inconclusive.

In a second set of experiments we stained *ephrinb1* mutants for pMLC at 60 hpf. Quantification of pMLC levels in the CBD by staining intensity revealed a significant increase of pMLC at the apical domain of up to 6-fold in *ephrinb1* mutants. Specifically, we have examined pMLC localization in the CBD in longitudinal sections and measured pixel intensities of the apical domain in multiple CBD positions per embryo. This suggests that EphrinB1 could directly control pMLC localization to the apical side, or indirectly for instance due to the altered cell shapes.

In the course of these experiments we have used a different and consistent staining protocol, and therefore replaced the previous panels; both protocol and antibody source are described in the Materials and Methods section.

Other Points

1 - In the introduction, a sentence between lines 50-51 can be moved to line 45 after "HPD", to indicate that the structure detailed in the introduction refers to fish. There are differences between mammalian and zebra fish HPD.

To clarify this we included a sentence describing the differences in the nomenclature of the EHD between zebrafish and mammals.

*2 - Instead of 2F11 in Fig 1 legend and line 212, use Annexin A4. It is unnecessarily confusing. Define *XlaEef1a1:GFPs854* for the readers.*

We changed 2F11 to Anxa4 (Annexin A4) consistently throughout the manuscript, and defined *XlaEef1a1:GFP^{s854}* as pan-endodermally expressed transgene in line 117 and legend to Figure1.

Reviewers' Comments:

Reviewer #1:

Remarks to the Author:

The authors have well addressed all my comments. I just have a minor comment as below.

I assumed that Eph/Ephrin signaling positively controls actomyosin contractility, thereby HPD morphogenesis. However, new data show that pMLC level was increased in ephrinb1 mutants (Fig. 4h). Does the increased pMLC level suggest hyper-contractility? Please explain what the increase mean in terms of actomyosin contractility and how the increased pMLC level leads to the defects in HPD morphogenesis. Is a similar increase in pMLC level expected in ephrinb2a or ephb3b mutants, as observed in ephrinb1 mutants?

Reviewer #2:

Remarks to the Author:

Review of the revised manuscript "A morphogenetic EphB/EphrinB code controls hepatopancreatic duct formation » by Thestrup et al.

This revised version of the article shows improved visualization of the HPD by 3D rendering through videos, more rigorous quantitative analyses (as it has been asked by several reviewers) and more mechanistic clues about EphB/EphrinB function in the HPD. Overall, the study now presents a set high quality data that nicely describe the mode of HPD morphogenesis and identify a novel role for EphB/EphrinB in this process.

There is no further improvement to be done.

Reviewer #4:

Remarks to the Author:

In this revised manuscript, the authors addressed most comments by this reviewers satisfactorily, including clarification of several points of confusion in writing and data interpretation as well as the demonstration of pMLC at the apical domain in ephrinb1 mutants. The studies providing a large body of evidence supporting a role for the EphB/Ephrin B system in hepatopancreatic duct formation, which may open the door for further studies on this important morphogenesis process.

Point by point response to editorial and reviewers' comments:

We thank the reviewers for their positive feedback and constructive suggestions for the discussion, which have helped us to further improve the manuscript. The revised manuscript now includes an extended description on the link between EphrinB1 and MyosinII contractility in CBD tubulogenesis, including a discussion of the potential functions of other EphrinBs and EphBs.

Reviewer #1:

I assumed that Eph/Ephrin signaling positively controls actomyosin contractility, thereby HPD morphogenesis. However, new data show that pMLC level was increased in ephrinb1 mutants (Fig. 4h). Does the increased pMLC level suggest hyper-contractility? Please explain what the increase mean in terms of actomyosin contractility and how the increased pMLC level leads to the defects in HPD morphogenesis. Is a similar increase in pMLC level expected in ephrinb2a or ephb3b mutants, as observed in ephrinb1 mutants?

An increase of pMLC levels at apical CBD membranes in *ephrinb1* mutants may seem surprising. However, the reduced length of apical membranes together with increased pMLC levels suggests hyper-contraction of the apical part of the cell. This is accompanied by an increased cell height and CBD shortening, suggesting that a balance between apical lengthening and lateral membrane shortening controls CBD epithelial rearrangement, tube differentiation and elongation. In wild type CBD, moderate pMLC levels at the apical membrane therefore prevent cell spreading on the one hand and hyper-contraction on the other. Hence, we propose that myosinII contractility changes dynamically at apical, lateral and basal membranes as tubulogenesis advances, mediating the specialised cellular behaviours of cell intercalation, lumen fusion and tube elongation. Our data demonstrates that EphrinB1 controls moderate pMLC levels, although whether directly or indirectly needs to be determined. The partly additive functions of EphrinB1 and EphrinB2a in the CBD would indicate that EphrinB2a may control pMLC levels in a similar fashion. EphB3b may act similarly on pMLC levels, however, milder cell shape CBD alterations at 60 hpf, point to additional requirements, e.g. pMLC regulation at earlier or later time points.

Reviewer #2:

No response required.

Reviewer #4:

No response required.